# Intervention targeted at physicians' treatment of musculoskeletal disorders and sickness certification: an interrupted time series analysis

Johanna Kausto ,[1] Tom Henrik Rosenström,[2] Jenni Ervasti ,[1] Olli Pietiläinen,[3] Leena Kaila-Kangas,[1] Ossi Rahkonen ,[3] Jaakko Harkko ,[4] Ari Väänänen,[1] Anne Kouvonen ,[4,5] Tea Lallukka [3]

[1]Finnish Institute of Occupational Health, Helsinki, Finland
[2]Department of Psychology and Logopedics, University of Helsinki Faculty of Medicine, Helsinki, Finland
[3]Department of Public Health, University of Helsinki, Helsinki, Finland
[4]Faculty of Social Sciences, University of Helsinki, Helsinki, Finland
[5]Administrative Data Research Centre, Queen's University Belfast, Belfast, UK

**Correspondence to**
Dr Johanna Kausto;
johanna.kausto@ttl.fi

## ABSTRACT

**Objective** An intervention was carried out at the occupational healthcare services (OHS) of the City of Helsinki beginning in 2016. We investigated the association between the intervention and employee sick leaves using interrupted time series analysis.

**Design** Register-based cohort study with a quasi-experimental study design.

**Setting** Employees of the City of Helsinki.

**Participants** We analysed individual-level register-based data on all employees who were employed by the city for any length of time between 2013 and 2018 (a total 86 970 employees and 3 014 075 sick leave days). Sick leave days and periods that were OHS-based constituted the intervention time series and the rest of the sick leave days and periods contributed to the comparison time series.

**Intervention** Recommendations provided to physicians on managing pain and prescribing sick leave for low back, shoulder and elbow pain.

**Outcome measures** Number of sick leave days per month and sick leave periods per year.

**Results** For all sick leave days prescribed at OHS, there was no immediate change in sick leave days, whereas a gradual change showing decreasing number of OHS-based sick leave days was detected. On average, the intervention was estimated to have saved 2.5 sick leave days per year per employee. For other sick leave days, there was an immediate increase in the level of sick leave days after the intervention and a subsequent gradual trend showing decreasing number of sick leave days.

**Conclusions** The intervention may have reduced employee sick leaves and therefore it is possible that it had led to direct cost savings. However, further evidence for causal inferences is needed.

## INTRODUCTION

Musculoskeletal disorders (MSD) are highly prevalent among the working-age population and result in extensive disability costs both for the individual and the society. In Finland in 2020, 26% of the days of sick leave periods lasting at least 11 calendar days were due to MSD.[1] Thus, sick leave is frequently granted

### Strengths and limitations of this study

► This study presents novel information on the association between introducing recommendations for management of pain and prescribing sick leave for musculoskeletal disorders and employee sick leaves.
► Comprehensive individual-level data from employer registers were linked to individual-level data from national registers.
► We used interrupted time series analysis with a comparison time series, which is a strong quasi-experimental design recommended for examining population-level interventions.
► Intervention status was not obtainable for the entire follow-up time and information on diagnoses was available only for sick leave periods that lasted over 11 calendar days.
► As all employees could be prescribed sick leave both at the occupational healthcare service and elsewhere, the validity of the comparison group could be debated; however, only the professionals in the intervention group were exposed to the intervention and the time series for the intervention and comparison groups were analysed separately.

for musculoskeletal pain symptoms, such as back, shoulder or elbow pain. However, scientific evidence does not support the beneficiality of long sick leaves. For example, it has been shown that for low back pain, from a point of view of recovery, staying active is better than bed-rest.[2 3]

Advice from primary care doctors can have a major impact on whether an individual takes sick leave and for how long.[4] A follow-up study found that visiting a medical specialist for a MSD was associated with a delayed full return to work irrespective of the severity of ailment.[5] Furthermore, earlier studies have found that sick-listing (sick leave prescribing) practices vary a lot across physicians.[6 7] A

Finnish study showed that differences in the length of prescribed sick leaves can be up to eightfold.[8] In line with this, a survey found that Finnish physicians needed more information, guidance and training on sick-listing.[9]

Against this background, an intervention was carried out at Occupational Health Helsinki, which provides occupational healthcare services (OHS) for the City of Helsinki employees. The intervention was launched in 2016. Introducing and increasing the use of shared guidelines on the management of pain and work disability and prescribing sick leave for low back, shoulder and elbow pain was the central element of the intervention. To investigate the association of this intervention with employee sick leaves, we carried out prospective controlled time series and binomial regression analyses using register-based data.

## METHODS
### Study design and population
Data were collected as part of the Helsinki Health Study.[10] The study population includes a sample of employees of the City of Helsinki. Women represent over 70% of the employees who work in general local administration, social and healthcare, education and culture, public transport, and technical services. All employees share the same personnel administration, policies and OHS.[10] Employees can use the services (including primary care services for other than work-related conditions) of their OHS provider, Occupational Health Helsinki, or they can seek treatment from public or private healthcare. The employees of the City of Helsinki can use self-certification for sick leaves lasting up to 5 days. In the case of self-certification, a medical certificate is not needed, but the employee notifies the supervisor when taken ill. In 2016, self-certification was mostly used among young employees (under 30 years of age).[11]

For the purposes of this study, individual-level register-based data on all employees who were employed by the city for any length of time between 2013 and 2018 (N=86 970) were analysed. Background information on participants was drawn from employer registers. Information on all sick leave periods (start and end dates and diagnostic codes) International Classification of Diseases, 10th revision (ICD-10) during this time was derived from employer registers and the national sickness insurance register of the Social Insurance Institution of Finland (SII). ICD-10 codes for sick leave periods were available from the SII for sick leave periods lasting more than 11 calendar days. Data on chronic somatic and psychiatric illnesses and purchases of prescribed psychotropic medication were obtained from the Register of Special Reimbursement for Medication Purchases (SII). Data from these three registers were linked based on unique individual number of the participants. These numbers are assigned to each citizen of Finland at birth and for migrants when they get a residence permit, and these are common to all administrative registers, enabling extensive record linkages.[12]

The study follows the Helsinki Health Study protocol in line with the University of Helsinki's guidelines and the European Union (EU) and Finnish data legislation. The City of Helsinki and register holders have given permission for data linkage.

### Patient and public involvement
As this is a register study, no patients were directly involved.

### Study context
#### Occupational Healthcare in Finland
In the Finnish OHS system, all employees are entitled to statutory preventive OHS provided by the employer. Most employers offer state-subsidised free primary care services for their employees on top of the statutory preventive OHS. OHS providers in Finland provide primary care services to employees in a way that is comparable to primary care in international comparisons in substance, but differs in that the service is targeted only to the working population.[13] The statutory tasks of OHS include assessment of the health and safety aspects of work, assessment and monitoring of employees' health and work ability, making initiatives for improvement and monitoring their implementation, advice and guidance, monitoring employees with disabilities and referring them to rehabilitation, cooperation with representatives of other healthcare services and social insurance, participation in organising first aid at the workplace, participation in activities that maintain work ability, and monitoring the quality and impact of occupational healthcare activities.[14]

#### Intervention
Occupational Health Helsinki provides occupational healthcare to almost 40 000 employees working annually in the city departments. The employees represent hundreds of different occupations, and a rather large supply of healthcare services supporting employees' work ability and the functioning of work units are offered. Occupational Health Helsinki staff consists of 150 healthcare professionals.

The aims of the intervention were to increase the quality and effects of occupational healthcare, to increase the use of mutually created and shared guidelines based on available scientific evidence in assessing work disability, to provide a practical instrument for physicians in managing pain and prescribing sick leave in the form of recommendations, to help physicians to take into account the complexity of the association between pain and work disability, and to provide up-to-date information to the patient about their pain condition, its treatment and how it may affect work and work ability. The goal was to reduce employee sick leaves and improve work ability.

Recommendations on work disability management and prescribing sick leave for low back pain, shoulder pain and elbow pain were launched first in January 2016 and they were the central and most noticeable component of the intervention. Recommendations were launched

in the form of three guideline papers (one for each diagnostic category). Training regarding the implementation of the guidelines was organised at OHS for physicians and all other relevant healthcare professionals, such as physiotherapists and nurses. Short educational sessions were offered, and an e-learning course and some coaching sessions were led by a pain specialist. An intensive follow-up of the sick leave trends was started at OHS. For instance, recent trends on sick leave were reported in the monthly meetings for the physicians. All new relevant OHS employees were briefed about the guidelines.

As an example, in the case of low back pain, the recommendation (online supplemental appendix 1) emphasised that the condition is very common and that nearly all people experience a disabling low back pain (ie, pain, ache and stiffness in the lower back area) at some point in their lives. The condition is usually benign in nature and heals spontaneously in a few days, and 90% of patients will fully recover in 4–6 weeks. Still, possible serious causes of back pain should be ruled out. In sick-listing, the following points should be considered: (1) in many cases sick leave is not needed; (2) there is no indication that physically strenuous work would slow down recovery; (3) if sick-listing is necessary, a few days up to a week should be sufficient in light duties; (4) if work includes heavy lifting, bending or rotation, 2 weeks of full-time sick leave may be needed; (5) factors that provoke pain should be temporarily modified; (6) and factors that suggest an increased risk of prolonged back pain and work disability include sleeping problems, depression and anxiety, fear avoidance beliefs, multisite pain, and problems at the workplace and dissatisfaction with work. The recommendations included a checklist for the physicians to be used at the appointment.

### Intervention status

Sick leave periods that were OHS-based (if a sick leave was preceded by a likely related appointment at OHS within 11 days, it was approximated that sick-listing had taken place at OHS) constituted the intervention time series, and the rest of the sick leave periods (no appointment at OHS, but appointment and sick-listing had taken place elsewhere, or the employee had used self-certification, in which case there had been no medical appointment at all) contributed to the comparison time series. The available registers contained information on appointments at OHS until 20 April 2017.

### Outcome

We calculated the number of sick leave days per month and sick leave periods per year.

### Covariates

Covariates that were available in the registers and were regarded as potential confounders included age, sex, occupational class (based on occupational title and categorised as upper grade non-manual employees, intermediate grade non-manual employees, lower grade non-manual

employees and manual workers), job contract (permanent, temporary) valid on 1 January 2016, chronic somatic illnesses (derived from the Register of Special Reimbursement for Medication Purchases and included diabetes, heart disease, rheumatoid arthritis, chronic asthma, stage 2 hypertension, Parkinson's disease, epilepsy, uraemia, bowel disease, multiple sclerosis disease and diseases of the pancreas, and categorised as 0=no or 1=yes) valid on 1 January 2016, and purchases of prescribed psychotropic medication (Anatomical Therapeutic Chemical classification N06 or N05, categorised as 0=no or 1=yes) between 1 November 2015 and 1 November 2016.

### Statistical analyses

We used an interrupted time series (ITS) design with a comparison time series.[15–17] The observational unit in the analyses was a sick leave day or period. The ITS analysis uses time series (a continuous sequence of observations at the population level taken repeatedly at equal intervals over time) for a specific outcome to establish an underlying trend which is 'interrupted' by an intervention at a specific point of time. A strength of ITS analyses is that they are generally not affected by typical confounding factors (such as age distribution or socioeconomic status) as they model a time trend in a context where the population composition (in terms of the confounding factors) remains rather constant before versus after the intervention. Measured time-varying confounders can be controlled for in the regression models. Unmeasured or unknown time-varying confounders can be controlled by adding a comparison group (not exposed to the intervention).[15] It has been recommended that the covariate balance between the intervention and comparison time series be explored, although it is not a prerequisite for the analysis.[16]

Our integer-valued (count or proportion) outcome variable was days of sick leave per month (range 0–30, or from 0 to days exposed within the 30-day period, if not with the employer for the full 30-day period), recorded separately for each month. Although each employee was followed up for multiple months, different employees may have different time average rate of sick leave days, introducing statistical dependencies to the observations. As our interests pertained to a population-level association, within-employee clustering in the outcome was considered a nuisance variable and its associations were modelled using generalised estimating equations.[18] Given the data characteristics above, the outcome was modelled using generalised (binomial) linear regression model[19] and the models were estimated with generalised estimating equations. Although our data were in the form of monthly counts, an exponentiated binomial regression coefficient can be interpreted as an OR for a sick leave day (vs no sick leave) for a randomly chosen day, adjusting for the other covariates in the model. As a sensitivity analysis, we also tested quasi-binomial

generalised linear model with usual maximum likelihood estimation, which gave similar results with narrower CI.

The main test of our hypothesis pertained to the following model equation:

$$g^{-1}(Y) = \beta_0 + \beta_{slope}(m - m_0) + \\ \beta_{immediate}X + \beta_{gradual}X(m - m_0)$$

where $Y$ is the outcome, $g$ is the logit link function, $X$ is 1 for all months after the intervention launch date (1 January 2016) and 0 before it, and $(m - m_0)$ is time (months, centred around the time of intervention,[20] $m_0$). Here, $\beta_{slope}$ is the linear trend estimate before the intervention, whereas $\beta_{slope} + \beta_{gradual}$ is the trend after the intervention took place. The intercept before the intervention is $\beta_0$ and after the intervention $\beta_0 + \beta_{immediate}$. Thus, $\beta_{immediate}$ is the immediate trend after the intervention and $\beta_{gradual}$ is the gradual trend after the intervention that accumulates at the rate $\beta_{gradual}$ per month after the intervention (for the remaining observation period). Besides this usual ITS setting, we took into account the annual fluctuation of sick leave by modelling an annual trend estimating the coefficients for two additional covariates with values $\sin(m2\pi/12)$ and $\cos(m2\pi/12)$, where the factor $2\pi/12$ scales monthly data points $m$ to the annual cycles.[21]

Because our outcome data timepoints (months) could include sick leave days from prescribed at OHS and other sources, defining a covariate for intervention group membership would have required us to prioritise either OHS or other sources. It is typical in controlled interrupted time series (CITS) design to include a binary covariate that gets a value of 1 when a unit belongs to the intervention group and a value of 0 when in the control group. In this study, participants had sick leaves both from OHS and non-OHS sources. Instead of formal testing of such covariate, we conducted a separate ITS design and analyses for the intervention and comparison groups. Both crude models and models adjusted for the covariates were fitted. Analyses were run separately for men and women, for short (≤11 calendar days) and longer (>11 calendar days) sick leave periods, and for sick leave periods (≥11 calendar days) in the diagnostic categories of back pain (ICD-10 M54.5), shoulder pain (ICD-10 M75) and elbow pain (ICD-10 M77.1).

As the sick leave recommendations potentially were associated with both the number of sick leave days and periods, the number of consecutive sick leave periods was computed within the index month plus 5 immediately preceding months. Monthly sample averages of the number of periods were investigated with otherwise similar linear models as in above but using a 6-month moving-average error structure. The 'arima' function of R V.3.5.1 (2018-07-02) was used. Local regression lines for the figures were drawn using loess function of 'stats' in R package (V.3.5.1), with the default options. The same software was used for all analyses.

## RESULTS

### Descriptive results

Our data included information on sick leave of a total of 86 970 employees covering more than 73 months. Due to workforce turnover, the monthly sample size of employees averaged at 41 289, with a minor variance across months (minimum 37 567, maximum 45 520, SD 2054). Overall, 70% of the employees were on sick leave at least once during the follow-up and 33% of the employees were prescribed sick leave at OHS at least once between 1 January 2013 and 21 April 2017. Of the employees who were prescribed sick leave at OHS, 95% were also prescribed sick leave elsewhere (or they used self-certification). Of those who were prescribed sick leave elsewhere (or used self-certification) at least once, 46% were also prescribed sick leave at OHS. Of all employees, 30% were not on sick leave at all. On average, employees were prescribed 0.36 days of sick leave per month at OHS and 0.82 sick leave days per month were prescribed elsewhere or were self-certified. The respective mean numbers of sick leave periods per employee per year were 0.49 and 1.96.

The descriptive statistics for all employees, employees with sick leaves prescribed at OHS and employees with other sick leaves (prescribed elsewhere or self-certificated) are shown in tables 1 and 2. Employees who were prescribed sick leave at OHS at least once by 21 April 2017 were more often permanently employed, older, female, and intermediate-level or lower-level non-manual workers (table 1).

### Monthly averages of the number of all sick leave days

Since it is generally recommended to draw simple plots of ITS analyses before entering into more complex model fitting,[15 16] we started by examining the monthly average days of sick leave per employee. Based on a visual inspection of monthly averages, the number of all sick leave days prescribed at OHS declined shortly after the intervention had started in 2016 (figure 1A), and similar temporal changes were not observed for other sick leave periods (figure 1B). After 21 April 2017, our register data could not differentiate between sick leave days that were OHS-based and other sick leave days, but we had a longer stretch of data when counting in all sick leaves. A decline in sick leave days after the intervention was seen on visual inspection of all sick leaves (thick line for local regression smoother), as well as an annual periodic variation (thin dashed line in figure 1C). When removing cyclic annual variation and a linear trend, visual suggestion of an association between the intervention and sick leaves remained. However, sick leaves appeared to return to preceding rates towards the end of the longest available follow-up data (in 2018) (figure 1D). It should be noted that the panels in

**Table 1** Descriptive statistics for all employees, employees with sick leaves prescribed at OHS and employees with sick leave prescribed elsewhere or self-certified (other sick leaves)*

| Characteristics | Sick-listing at OHS (n=25 200) | Other sick leaves (n=52 174) | All (n=75 962) |
|---|---|---|---|
| Age, mean (SD) | 44.67 (12.32) | 42.16 (13.02) | 40.27 (14.08) |
| Men (%) | 20 | 23 | 27 |
| Occupational class (%) | | | |
| Upper grade non-manual employees | 21 | 24 | 27 |
| Intermediate grade non-manual employees | 27 | 25 | 22 |
| Lower grade non-manual employees | 37 | 36 | 34 |
| Manual workers | 16 | 15 | 17 |
| Job contract (%) | | | |
| Permanent | 82 | 67 | 54 |
| Prescribed reimbursed purchases of medication for chronic diseases (%) | | | |
| Yes | 21 | 17 | 16 |
| Prescribed reimbursed psychotropic medication (%) | | | |
| Yes | 12 | 9 | 8 |

*Note that many employees were prescribed sick leave both at OHS and elsewhere (or used self-certification). The group 'All' contains also employees with no sick leaves.
OHS, occupational healthcare services.

figures 1 and 2 are not comparable with each other as the y-axes are different. The figures illustrate how the trend in sick leave days changes at the time of the intervention rather than compare the absolute values. Due to relatively small change in trend, similar y-axis would not illustrate the trend.

Formal modelling (adjusting for seasonal variation) did not reveal an immediate diminishing trend ($\beta_{immediate}$ =0.01, 95% CI −0.052 to 0.072), but did detect a gradual diminishing trend ($\beta_{gradual}$=−0.007, 95% CI −0.013 to −0.001) in the number of OHS-based sick leave days. The corresponding numbers in the comparison group were similar in direction but not statistically significant ($\beta_{immediate}$=0.068, 95% CI −0.065 to 0.201; $\beta gradual$=−0.009, 95% CI −0.022 to 0.004) (data not shown). These formal findings would support a gradual diminishing trend after the intervention, but we were able to achieve a greater statistical power by accessing employee-level data instead of the above-discussed monthly averages.

### Results on employee-level data: number of all sick leave days

We modelled the number of sick leave days using binomial regression on monthly sick leave days per employee. Figure 2A,B illustrates the model predictions for the number of all sick leave days. For convenient illustration, we plotted the model estimates overlaid on the above-discussed monthly averages. These models used 2 160 445 observations from 75 962 individuals instead of 52 monthly averages. The exact binomial regression coefficients (with logit link function) as well as the OR for a sick leave day at index month versus the preceding month are presented in table 3.

We observed a clear gradual diminishing trend in the number of all-cause OHS-based sick leave days (model 1; table 3). A similar gradual trend was found in the comparison group (sick-listing elsewhere or self-certification), but this was offset by an increase in the level of the number of sick leave days at the time of

**Table 2** Monthly average number of sick leave days per month per employee by intervention status (2013–2017)

| | Sick-listing at OHS | | | Other sick leaves | | |
|---|---|---|---|---|---|---|
| | Average | Range | SD | Average | Range | SD |
| All sick leave days | 0.380 | 0–30 | 2.59 | 0.840 | 0–30 | 3.40 |
| All short sick leave periods* | 0.117 | 0–25 | 0.79 | 0.392 | 0–24 | 1.24 |
| Sick leave for low back pain† | 0.005 | 0–30 | 0.36 | 0.007 | 0–30 | 0.39 |
| Sick leave for shoulder pain† | 0.020 | 0–30 | 0.71 | 0.014 | 0–30 | 0.59 |
| Sick leave for elbow pain† | 0.004 | 0–30 | 0.28 | 0.002 | 0–30 | 0.19 |

*≤11 calendar days.
†>11 calendar days.
OHS, occupational healthcare services.

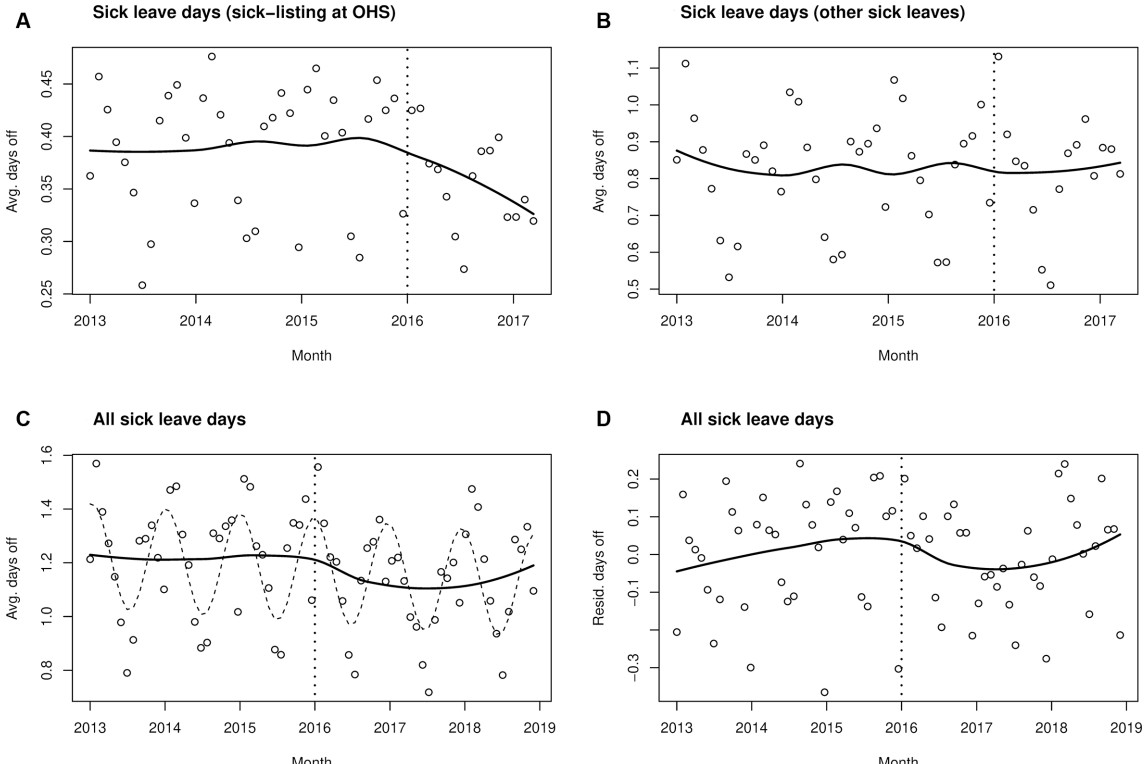

**Figure 1** Model fits. Monthly averages of the number of all sick leave days. Intervention status could be specified until 21 April 2017 (A and B). In C and D, the intervention status is not differentiated. Vertical dashed lines indicate the timepoint of intervention (A–D). Circles denote observed averages, thick lines their local regression fits (smoothed averages) (A–D) and thin dashed line annual periodic variation (C). Note the differing scales in the figures. OHS, occupational healthcare services; Avg, average; Resid, residual.

intervention. Adjusting for the covariates (model 2; table 3) did not affect the estimates significantly. We additionally ran the analyses separately for men and women (not shown in the table). The results were in line with the first model (total sample). The association between the intervention and sick leaves found in all-cause sick leave was not paralleled in the diagnostic group-specific findings (table 3).

While a gradual diminishing trend of sick leave days with an OR of 0.986 may appear small, we can illustrate the association between the intervention and sick leave days in a practical way as follows. Assuming employees on average have 1.2 sick leave days per month (cf, figure 1C), their baseline risk rate per day would be 0.04. Small risk ratios closely match the OR. Thus, if the other factors have already been adjusted for, the intercept for a logistic model would be $\beta_0 = \log(0.04)$. Then the gradual intervention estimate of $OR_{gradual} = 0.986$ would imply that the OR of a sick leave day 1 year after the intervention has reduced to $\exp\left(\beta_0 + 12 \times \log\left(OR_{gradual}\right)\right) \approx 0.034$ and the risk rate of a sick leave day to approximately 0.033. Thus, the gradual intervention estimate on average amounts to $(0.04 - 0.033) \times 30 \times 12 = 2.52$ of avoided sick leave days per year per employee. Expressed in another way, the intervention may have saved 2520 sick leave days per year per 1000 employees.

## Results on employee-level data: number of all sick leave periods

As a sensitivity analysis, we investigated the number of all-cause sick leave periods as an outcome. A sick leave period is a consecutive spell of sick leave days. We report the results in units of periods per year per employee, although we computed the periods per index month plus 5 preceding months (figure 2C,D). We analysed the monthly averages with a moving-average regression model. For all sick leave periods prescribed at OHS, we did not detect any immediate change in trend ($\beta_{immediate}$=0.017, 95% CI −0.011 to 0.046), but a gradual diminishing trend in the number of sick leave periods was found ($\beta_{gradual}$ =−0.009, 95% CI −0.013 to −0.005). The findings on the gradual change of trend were similar in the comparison group ($\beta_{gradual}$=−0.014, 95% CI −0.026 to −0.003), whereas the immediate change in trend indicated that an increase in sick leave periods occurred in the comparison group at the time of the intervention ($\beta_{immediate}$=0.141, 95% CI 0.067 to 0.214).

## DISCUSSION

In this register-based study among municipal employees, we analysed sick leave trends before and after an intervention targeted at physicians' management of musculoskeletal pain-related work disability and prescribing sick leave.

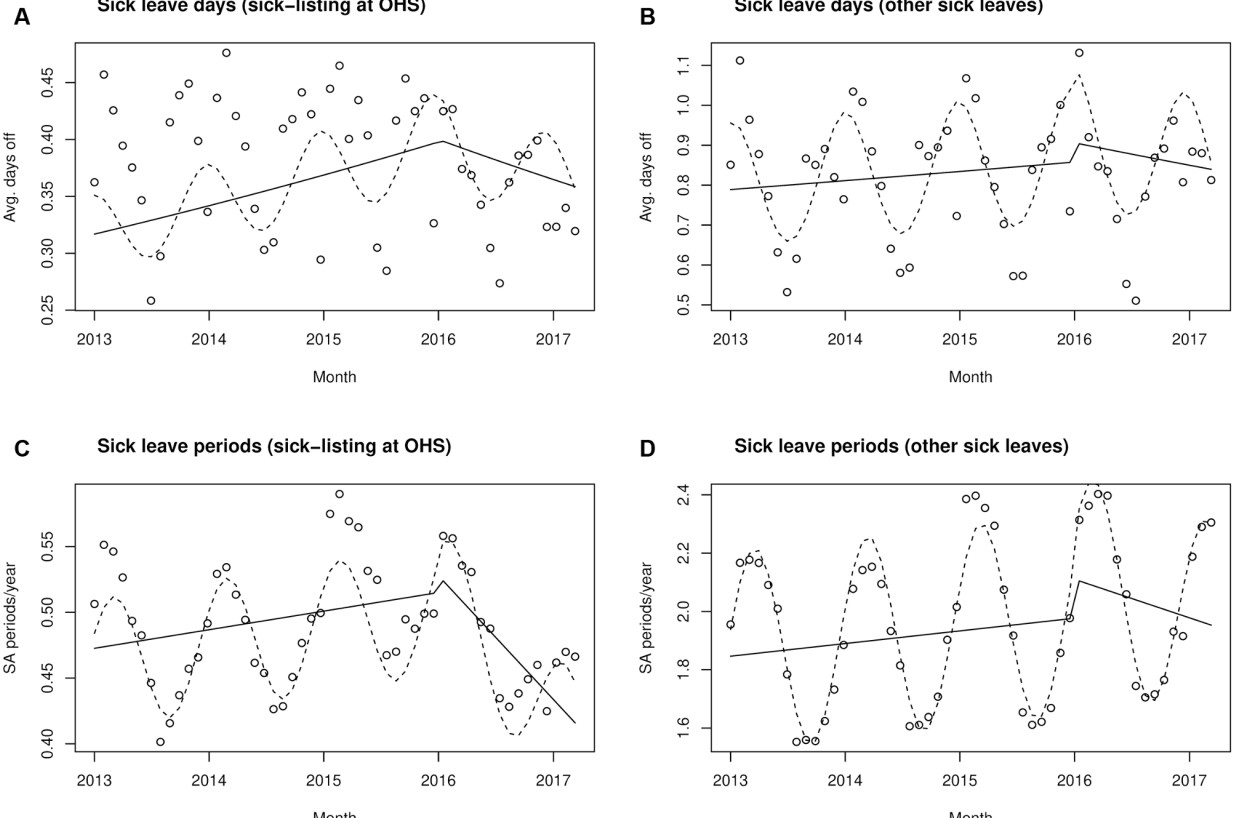

**Figure 2** Model fits. Monthly averages of the number of all sick leave days (A and B) and periods (C and D). Observed data points denote simple monthly averages and the lines represent GEE model fits to employee-level data. The dashed line represents the full model prediction, including annual variations, whereas the solid line is the ITS part of the model. Note the differing scales in the figures. GEE, generalised estimation equation; ITS, interrupted time series analysis; OHS, occupational healthcare; Avg, average; SA, sickness absence services.

The intervention was carried out in an OHS setting. We examined monthly averages of all-cause sick leave days and found no immediate change in trend among OHS-based sick leave days, but did detect a gradual decreasing trend in the number of OHS-based sick leave days by 2.5 days per year per employee. The corresponding findings were similar in direction but not statistically significant for the comparison time series (sick-listing elsewhere or self-certification). In the employee-level data (with a larger statistical power), we found a clear gradual decreasing trend in the number of OHS-based all-cause sick leave days. Similar gradual decreasing trend was found in the comparison time series, but this was offset by an increase in the level of sick leave days and periods at the time of the intervention. A hypothetical organisation with 40 000 employees and an average direct cost of €250 per sick leave day (numbers similar to those of the City of Helsinki)[22] is likely to have resulted in substantial annual savings at the time of the intervention.

The knowledge base on physicians' sick-listing practices is accumulating. The present study suggests that there was an association between the intervention and the sick-listing practices of physicians working at Occupational Health Helsinki. There was a downward trend in the number of OHS-based sick leave days and periods. Nevertheless, the findings may also suggest that soon after

the intervention was carried out at OHS, the employees increasingly sought treatment elsewhere (or used self-certification increasingly) as an increase in the level of other sick leaves was detected. This trend levelled down, however.

As far as we know, there are no previous studies investigating the effects of introducing guidelines on prescribing sick leaves (as compared with the situation where no guidelines exist). A few studies have compared different ways of implementation.[23 24] There are many previous studies on interventions attempting to change physicians' behaviour in some other areas of clinical practice. Findings show that success in changing the behaviour and adherence to different clinical guidelines vary. Some studies reported success in behaviour change,[25 26] whereas other studies reported limited success or found the interventions ineffective.[27–29] Moreover, a change in physicians' behaviour does not automatically reflect in patient outcomes.[28] Guidelines have been found to be a necessary but insufficient step in changing clinical care.[30] The importance of active implementation of guidelines and efforts to include the new practice into existing organisational procedures, which both were attempted in this intervention, has been emphasised to achieve sustainable results.[31 32] As the context of the intervention is important, the results of

**Table 3** Binomial regression models predicting the number of sick leave days

| | Sick-listing at OHS | | | | Other sick leaves | | | |
|---|---|---|---|---|---|---|---|---|
| | β coefficient | SE | OR† | 95% CI | β coefficient | SE | OR† | 95% CI |
| **Sick leave days (all periods, all DC:s) (crude model) (n=2 160 445 employee-months, n=75 962 employees)** | | | | | | | | |
| Immediate change in trend | 0.012 | 0.035 | 1.012 | 0.95 to 1.08 | 0.060** | 0.018 | 1.062 | 1.03 to 1.10 |
| Gradual change in trend | −0.014*** | 0.003 | 0.986 | 0.98 to 0.99 | −0.008*** | 0.002 | 0.992 | 0.99 to 1.00 |
| Linear trend before intervention (slope) | 0.006*** | 0.001 | 1.006 | 1.00 to 1.01 | 0.002*** | 0.001 | 1.002 | 1.00 to 1.00 |
| **Sick leave days (all periods, all DC:s) (fully adjusted model) (n=2 148 566 employee-months, n=74 701 employees)** | | | | | | | | |
| Immediate change in trend | −0.003 | 0.032 | 0.997 | 0.94 to 1.06 | 0.057** | 0.018 | 1.059 | 1.02 to 1.10 |
| Gradual change in trend | −0.010** | 0.003 | 0.990 | 0.98 to 1.00 | −0.007*** | 0.002 | 0.993 | 0.99 to 0.99 |
| Linear trend before intervention (slope) | 0.007*** | 0.001 | 1.007 | 1.01 to 1.01 | 0.003*** | 0.001 | 1.003 | 1.00 to 1.01 |
| **Sick leave days (short periods, <11 calendar days, all DC:s) (n=2 160 445 employee-months, n=75 962 employees)** | | | | | | | | |
| Immediate change in trend | 0.105*** | 0.024 | 1.111 | 1.06 to 1.16 | 0.196*** | 0.010 | 1.217 | 1.19 to 1.24 |
| Gradual change in trend | −0.020*** | 0.002 | 0.980 | 0.98 to 0.98 | −0.009*** | 0.001 | 0.991 | 0.99 to 0.99 |
| Linear trend before intervention (slope) | 0.001 | 0.001 | 1.001 | 1.00 to 1.00 | −0.003*** | <0.001 | 0.997 | 1.00 to 1.00 |
| **Sick leave days in low back pain (periods ≥11 calendar days) (n=2 160 445 employee-months, n=75 962 employees)** | | | | | | | | |
| Immediate change in trend | −0.547 | 0.340 | 0.579 | 0.30 to 1.13 | −0.205 | 0.283 | 0.815 | 0.47 to 1.42 |
| Gradual change in trend | 0.040 | 0.035 | 1.041 | 0.97 to 1.12 | −0.019 | 0.031 | 0.981 | 0.92 to 1.04 |
| Linear trend before intervention (slope) | −0.004 | 0.009 | 0.996 | 0.98 to 1.01 | −0.004 | 0.007 | 0.996 | 0.98 to 1.01 |
| **Sick leave days for shoulder pain (periods ≥11 calendar days) (n=2 160 445 employee-months, n=75 962 employees)** | | | | | | | | |
| Immediate change in trend | 0.365 | 0.207 | 1.441 | 0.96 to 2.16 | −0.426 | 0.292 | 0.653 | 0.34 to 1.16 |
| Gradual change in trend | −0.023 | 0.023 | 0.977 | 0.93 to 1.02 | 0.013 | 0.025 | 1.013 | 0.97 to 1.06 |
| Linear trend before intervention (slope) | −0.004 | 0.008 | 0.996 | 0.98 to 1.01 | −0.003 | 0.008 | 0.997 | 0.98 to 1.01 |
| **Sick leave days for elbow pain (periods ≥11 calendar days) (n=2 160 445 employee-months, n=75 962 employees)** | | | | | | | | |
| Immediate change in trend | −0.964 | 0.514 | 0.381 | 0.14 to 1.04 | −0.398 | 0.486 | 0.672 | 0.26 to 1.74 |
| Gradual change in trend | −0.034 | 0.072 | 0.967 | 0.84 to 1.11 | −0.079 | 0.084 | 0.924 | 0.78 to 1.09 |
| Linear trend before intervention (slope) | 0.003 | 0.014 | 1.003 | 0.98 to 1.03 | −0.001 | 0.018 | 0.999 | 0.96 to 1.04 |

Adjusted for age, sex, occupational class, job contract, prescribed reimbursed purchases of medication for chronic diseases and prescribed reimbursed psychotropic medication.

Follow-up from 1st January 2013 to 20th April 2017.

*p<0.05, **p<0.01, ***p<0.001.

†Immediate intervention estimate: OR was estimated for any employee to have any sick leave day after versus before the intervention. Gradual intervention estimate: OR was estimated for any employee to have a sick leave day at index month versus preceding month during the period after the intervention.

DC, diagnostic code; OHS, occupational healthcare services.

an intervention in one environment cannot directly be applied to another context.

A strength of this study is its design based on register data on all employees. We could link comprehensive data from employer registers to data from national registers using personal identification numbers, resulting in no loss to follow-up. The use of administrative registers in research always entails limitations as they are not typically collected for research purposes. Nevertheless, in general, the quality of registers in Finland, as in other Nordic countries, is good and administrative registers offer unique opportunities and are frequently used in research.

Applied ITS analysis with a comparison time series has been recommended for analysing population-level interventions.[15 33] A limitation was that the intervention status was approximated (an OHS-based sick leave was preceded by a likely relevant appointment at OHS) and the information was not available for the entire follow-up period. The validity of the comparison group can be debated[17] as all employees could be prescribed sick leave both at the OHS and elsewhere (or use self-certification in short sick leaves). However, only OHS professionals were exposed to the intervention and the time series for the intervention and comparison groups were analysed separately. Thus, our analysis should be thought of as being in between ITS and CITS in what comes to scientific value of evidence.

Because the intervention was targeted at managing pain and prescribing sick leave from the first day of work disability, a limitation was that we had access to diagnostic data only for sick leave periods that lasted over 11 calendar days. This, together with the low base rate for specific diagnoses, may explain the result where the association between the intervention and sick leaves found in all-cause sick leaves was not replicated in diagnosis-specific findings of sick leaves (lasting over 11 calendar days) due to low back, shoulder and elbow pain. MSDs constitute a major cause of sick leave. The fact that an association between the intervention and sick leaves was seen among all-cause sick leaves (regardless of the length of the sick leave period) may also indicate that recommendations given for these three specific disorders influenced how physicians managed work disability and prescribed sick leave altogether. Unfortunately, we did not have information on part-time sick leaves, and for this reason we could not take this into account in the analyses. In Finland, partial sickness benefit can be granted only after 11 days of full-time sickness absence. Partial sickness benefit days comprised only 7% of all compensated sickness benefit days in Finland in 2016.

The recommendations for physicians constituted chronologically the first and the main element of the intervention. Direct access to physiotherapists without a doctor's referral was introduced in the following year, and thereafter it was not possible to differentiate the association between these different elements and sick leaves.

To conclude, we observed a gradual declining trend in the number of sick leave days and periods prescribed at OHS after the intervention. This is consistent with the intervention being effective; however, the causality of the relationship is unclear due to similar findings in the comparison group. The estimated reduction of 2.5 sick leave days per year per employee could leave to substantial savings for a large employer such as the City of Helsinki. This can be considered an economically relevant intervention, but further evidence for its causal interpretation is needed.

**Correction notice** This article has been corrected since it was first published. Table 3 footnote has been updated.

**Acknowledgements** We would like to thank Eevamaija Tuovinen MD (chief medical officer), Helena Miranda MD, PhD, and Tiina Pohjonen PhD (CEO) of Occupational Health Helsinki for providing information on the intervention and its context.

**Contributors** Conceptualisation and methodology: JK, THR, OP, AV, JE, OR, JH, AK and TL. Writing - original draft preparation: JK and THR. Writing - review and editing: JK, THR, JE, OP, LK-K, OR, JH, AV, AK and TL. Guarantor and project administrator: JK. All authors have read and agreed to the published version of the manuscript.

**Funding** This work was supported by the Social Insurance Institution of Finland (grant #9/26/2019; JK, THR, JE, AV and LK-K), the Finnish Work Environment Fund (grant #190081; JK, THR, JE, AV and LK-K), the Academy of Finland (grants #1294514 and #319200; OR and TL) and the Economic and Social Research Council (ESRC) (grant #ES/S00744X/1; AK).

**Competing interests** None declared.

**Patient consent for publication** Not required.

**Ethics approval** The ethics committees of the Department of Public Health, the University of Helsinki and the health authorities of the City of Helsinki approved the Helsinki Health Study protocol. The ethical approval applies to the current study.

**Provenance and peer review** Not commissioned; externally peer reviewed.

**Data availability statement** All data relevant to the study are included in the article or uploaded as supplementary information. No additional data are available.

**ORCID iDs**
Johanna Kausto http://orcid.org/0000-0002-2898-0018
Jenni Ervasti http://orcid.org/0000-0001-9113-2428
Ossi Rahkonen http://orcid.org/0000-0002-7202-3274
Jaakko Harkko http://orcid.org/0000-0001-8682-1544
Anne Kouvonen http://orcid.org/0000-0001-6997-8312
Tea Lallukka http://orcid.org/0000-0003-3841-3129

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
