## [Reviewer comments · BMJ Open]

ARTICLE DETAILS

TITLE (PROVISIONAL)	An intervention targeted at physicians' treatment of musculoskeletal disorders and sickness certification: An interrupted time series analysis
AUTHORS	Kausto, Johanna; Rosenström, Tom Henrik; Ervasti, Jenni; Pietiläinen, Olli; Kaila-Kangas, Leena; Rahkonen, Ossi; Harkko, Jaakko; Vaananen, Ari; Kouvonen, Anne; Lallukka, Tea

VERSION 1 – REVIEW

REVIEWER	Bengtsson Bostrom, Kristina R&D Centre Skaraborg Primary Care
REVIEW RETURNED	17-Nov-2020

GENERAL COMMENTS	Ms bmjopen-2020-047018 entitled "The effectiveness of an intervention at physicians' treatment of musculoskeletal disorders and sickness certification: An interrupted time series analysis". Comments to the Review checklist 3. The study design is not appropriate to study the effectiveness of the intervention i.e. to study causality as also the authors point out. Using register data not intended for use in research is a special challenge and this should be reflected in the description of the results. Further, there is a risk that individuals beginning in the intervention group, went over to the control group in order to get a sick-note. Finally, the controls are not appropriately described to ensure that they were comparable with the individuals in the intervention group. 4. The method especially of the intervention is not enough described. For instance, how the guidelines were implemented and iterated to newly employed staff. How other staff apart from physicians were involved in the intervention? This should be described in more detail. 7. The statistics is much elaborated. I am not an expert in this field so I recommend a statistician should look at this part. I think that there might be inherent weaknesses in the material that can't be compensated for by elaborated and complicated statistics. 10. The y-axis in the panels of figures (1 and 2) have different scales, which makes it difficult to compare the results in the intervention and control groups. The time scales are also different in figure 2 panels a and b compared to panel c and d. In these figures it seems to be a rise in sick days the succeeding years 2017-2019. These data is not commented.
--

	11. In the discussion the authors uses the word effect to describe the presumed outcome of the intervention. I suggest that a less decisive word is used here as I think there is a risk for bias in the data that make the conclusions uncertain. 15. The language needs revision, and very long sentences in several places in the manuscript should be shortened. Comments to the authors Studies of interventions with the aim to reduce sick leave are much longed for, but are also difficult to perform. The current study is from Finland where the authors have used register data to evaluate an intervention using guidelines on musculoskeletal disorders aimed at sick-listing physicians. I have several issues that need to be addressed. Title The word effectiveness gives the impression of causality which I think is to strong as there are many shortcomings of the data, which also the authors recognize. It could be an association worth to investigate under more strict conditions. The wording in the title, abstract and main text of the manuscript should be revised with this in mind. Introduction. Avoid references in Finnish language. For instance I recommend instead of reference #8 the authors might use: Kankaanpää AT, Franck JK, Tuominen RJ. Variations in primary care physicians' sick leave prescribing practices. Eur J Public Health. 2012;22:92-6. doi: 10.1093/eurpub/ckr031. PMID: 21441559? Also, reference #9 might be replaced by Hinkka K, Niemelä M, Autti-Rämö I, et al. Physicians' experiences with sickness absence certification in Finland. Scand J Public Health. 2019;47(8):859–866. Does reference #11 have an Abstract in English? I not, the results ought to be described in more detail in the text. Methods. The nature of the intervention is not enough described. How were the guidelines implemented and iterated to newly employed staff? What did the guidelines look like? Were they including diagnostic procedures, treatment options in addition to the recommended sick leave length? Were other staff apart from sick leave prescribing physicians involved in the intervention for instance physiotherapist, occupational therapists, or psychologists treating the patients? Were also the employers involved in the intervention? Especially the practical instrument used in this context (page 7, line 51) and the checklist described on page 9, line 3 should be described or attached in an appendix. The control group described on page 9, line 9 includes individuals with self-certification. I presume that the controls thus include individuals with common cold and other self-limiting conditions that could distort the results. Why didn't you exclude those with self-certification from the study? Outcome, page 9. Were any patients sick listed part-time? If so was this accounted for in the analysis? Covariates, page 9. Among diseases that were included as covariates in the analyses depressive disorders are not
--	--

mentioned. I would have expected that depression, bourn-out syndrome and other psychiatric conditions should be included. These are conditions associated with chronic pain and might make return to work difficult. Are psychotropic drugs a proxy for these conditions and if so were the underlying diagnoses validated and medication prescriptions occurring simultaneously with the sick leave periods?

Statistical analyses. The methods are extensively described. I suggest that a great part of the text, specifically on page 11, could omitted or moved to an appendix.

Page 12, line 10-19. I don't understand how the authors handled individuals with a mixture of sick leave certificates from OHS and other sources. Do I perhaps misunderstand or does this mean that a single individual could be in both groups, in separate time intervals? If so, how could the authors disentangle the effect of the intervention from the "control effect"? I suggest exclusion of individuals with a mix of certificates from OHS and control, especially if the "mix" occurred during a short interval of time.

Results.

Page 13, line 6, "87% of all employees were prescribed sick leave at least once" is in contrast to line 18; "Of all employees 30 % were not on sick leave at all". Please explain this discrepancy.

Data in Table 1 is unclear. It seems that 1,412 individuals, $(1,412 = (25,200 + 52,174) - 75,962)$ were prescribed sick leave both at OHS and by others i.e. they belong to both groups making the comparison between groups difficult. Further, how come that the mean age is lower in "All" compared to "sick listing at OHS" and "other sick leaves"? Is this because the 1,412 individuals are counted twice?

Figure 1-2. The scales on the x-axis are different in the panels describing the results in the intervention and control groups, which can distort the impression of change in sick leave days and periods. The presumed effect of the intervention also seems to disappear in later years (Figure 1) which should be discussed by the authors.

Page 13, lines 53 and following; the interpretation of the results belongs to the Discussion section.

Page 15, lines 53 and following. The calculation of the reductions in costs due to the estimated reduction in sick leave days in the intervention group is not supported by any observed data. In a defined population i.e. all employed in the City of Helsinki there ought to be data on reimbursement costs to support the calculations in this study. I suggest that the authors include such data.

Discussion

The authors state that there was "a clear gradual intervention effect decreasing the number of OHS-based all-cause sick leave days. Similar gradual decreasing trend was found in the control time series, but that was offset by an increase in the sick leave days and periods at the time of intervention". The sharp rise of sick leave days in the control group immediately after the intervention is surprising. As the author speculate it could be due to patients

	moving from the intervention group to the control group in order to be sick listed. This is a very plausible explanation in my view. Thus, the comparison between these groups might not be a valid measure of effects of the intervention. Especially the notion that the decline in sick leave days in the control group was off-set by the rise in the beginning. This could also be interpreted as an off-set to the decline in the intervention group (as patients from OHS could seek a second opinion of their need for sick leave in caregivers outside OHS). Somehow, the suspected spillover of patients from the intervention to the control group must be investigated in the data and accounted for in the analysis. Further, other data for instance changes in the costs (savings of 25 million Euros) for sick leave ought to be included to support the authors' conclusions. The authors acknowledge this situation, page 18, lines 34 and following. The conclusions should accordingly be much more cautious. Of course, as the authors state a prospective intervention study that would enable tracking the patients' sick-listing behavior could shed further light on the effectiveness of such an intervention.
--	--

REVIEWER	Mulimani, Priti University of Washington, Oral Health Sciences
REVIEW RETURNED	12-Feb-2021

GENERAL COMMENTS	As a health-care practitioner in an international setting, I have no exposure to the Finnish health system, but this paper for the most part does a pretty great job of explaining the functioning and administrative aspects underlying the Finnish health system study in a way that is understandable for international readers. For the most part this is a very well-planned, - executed and -written study. I had only the following few suggestions / question-  1. From reading the paper I inferred that "sick-listing" refers to the practice of recommending sick-leave to the patients by the health care provider. It would be helpful to have a definition/explanation of the term "sick-listing" in the beginning, for uninitiated readers. 2. The control time-series included patients with "no appointment at OHS, but appointment and sick-listing had taken place elsewhere or the employee had used self-certification, in which case there had been no medical appointment at all". How did the study account for self-medication(previously purchased, hence no current purchase registered in the pharmacy) or other self-interventions in this group which would have gone unreported? 3. In methodology please provide a reference for – "Analyses were run separately for men and women (who are well known to differ in sick leave behaviour)..." 4. Comparison with previous studies and their findings has been broadly summarized in a paragraph. I would prefer to see more granular details as to how specific findings from the current study fared with respect to the previous ones.
---

REVIEWER	Toyooka, Takeshi Nishikawa Orthopaedics Clinic, rehabilitation
REVIEW RETURNED	04-May-2021

GENERAL COMMENTS	Thank you for your great work. Although the article have important message, there are too much results in your article. Which factor will affect to the intervention effect? I think there are no need to show without table data. The content of the cost is in the results, but this should be moved to the discussion section. As you know, if you want to compare 4 group's average data, I think two-way ANOVA is more appropriate. Don't you need to compare "sick listing at OHS" group and "other sick leaves" group? Otherwise, would you divide the purpose into intervention effect and its duration?
---

REVIEWER	Reeves, David University of Manchester, Institute of Population Health; Centre for Biostatistics
REVIEW RETURNED	20-May-2021

GENERAL COMMENTS	This paper applies an interrupted time-series design to try and assess whether an intervention consisting of recommendations on pain management and sick leave prescribing for OHS physicians, reduced numbers of sick leave days taken by employees of the City of Helsinki, Finland over the subsequent one-year period. The paper is interesting and the analysis mostly sound, however there are some important weaknesses that need to be addressed before publication could be recommended, as described below. Major points P11 line 13. P10 indicates that the key outcome was an integer count of number of days of sick leave per month, but on p11 it is stated that this outcome was modelled using a "generalised (binomial) linear regression model" (GEE) and the model equation is stated to use the logit-link function. Binomial/logit is normally used for a dichotomous outcome (eg values of 0 or 1), not a count. Although such a GEE model may produce a solution (with the outcome expressed as a proportion) it is very unusual to select such a model in preference to a Poisson or negative binomial, which are specifically designed for count outcomes. No justification has been given for the choice of a binomial link, but a particular concern is that this may under-estimate variances relative to a Poisson or Negative Binomial model since the data looks to be very over-dispersed: from table 2, OHS sick days had a mean of 0.38 and SD of 2.59, suggesting that a very high percentage of the group had zero sick days. To inform readers, a graph of the sick leave distribution (separately for OHS and Other) needs to be provided. Some justification, supported by references, for using a binomial model is required. The concern is that multiple models may have been fitted and the binomial chosen on the grounds that it gave the most favourable results – a good justification is therefore required to dispel such a concern. In addition, because of the above concerns, sensitivity analysis is required to assess the robustness of the results against distributional assumptions. My recommendation would be to run at least two additional sensitivities: (1) using a Negative Binomial model; (2) using bootstrapped SEs that make no distributional assumptions. The
---

authors might also want to consider using a zero-inflated negative binomial model, though interpretation may get very complex in the context of interrupted time-series.

P12. No formal statistical comparison is made of the regression parameters for the OHS and control groups, even though this should be relatively easy by including both groups in the same regression and interacting the regression terms with a group identifier. Since a direct comparison would normally be expected, some justification for this decision needs to be given. Personally, I have some qualms over the control group as I am not convinced that they represent a reasonable counter-factual: first, table 1 shows that a much smaller proportion of controls were on permanent job contracts, and second – probably related – prior to the intervention, mean numbers of sick-leave days for controls were more than twice those for the OHS-group – this suggests that pre-intervention, OHS may have already been operating to a very different protocol compared to “other” services. Thus I would suggest that there are important systematic differences between the groups making a direct comparison problematic.

Early on in the paper the authors make a lot of the claim that they included a control group. However, controls are only informative to the degree that they are similar enough to the intervention group to make comparisons meaningful. In this study the controls occupy an awkward middle-ground: they are not similar enough to justify formal comparisons (in my opinion), but may provide useful contextual information. I think use of the term “controls” here is not advisable, since it implies a high degree of comparability. The authors should consider using the term “comparison group” instead, and give suitable caveats – such as those I have expressed above – as to why they should not be treated as direct controls.

P14 line 46. “We modelled the number of sick leave days using binomial regression. “Binomial regression” implies that the outcome was a binomial (i.e. could take 2 values only): this will be confusing to many readers. It gives the impression that the outcome has been dichotomised, when it has not.

In two respects the seasonal sin/cos wave is not a particularly good fit to some of the data. First, from Figure 1 and 2 it can be seen that there is a strong “Christmas effect” on sick leave, with the December(?) value dropping right down in just about every year – perhaps because many people were taking holiday? These points strongly diverge from the wave function. Second, the wave is a particularly poor fit to the ONS group (figure 2a), with the large majority of points in the pre-intervention period being very high above the wave. This suggests that the fit of the pre-intervention slope is highly biased, supported by the fact that it also deviates greatly from the moving average line in figure 1a. Over the pre-intervention period I counted 27 data points above the pre-intervention line of slope but only 10 below it: this is strongly suggestive of a bias. The fit is also poor to the post-intervention period. I note in passing that the same problem does not seem to affect the model fit to the control dataset (figure 2b), though the “Christmas effect” is still very evident.

Given this, I think it essential that the OHS group model be re-evaluated. If figure 2a is accurately drawn then something has

clearly gone wrong in the analysis. One option might be to test the sensitivity to the sin/cos wave assumption, by replacing it with calendar month (jan, feb, etc) as a categorical variable; this would also resolve the Christmas effect issue.

P16. The authors estimate the mean 12-month reduction in sick leave days resulting from the intervention. The calculation is mathematically correct. However, it is based on assuming that the observed trend prior to the intervention would have continued unchanged had the intervention not happened. This is where I have a real difficulty with the paper. The control time-series shows a similar (and significant) change in trend after the intervention. Comparisons are made complex by the “jump” in mean sick days for the control group at the start of 2016, but even allowing for this, the mean number of sick days for controls at the end of 2016 was no greater than at the start.

Thus on the evidence of the control group, it is not valid to assume that sick leave in the OHS group would have continued along the pre-intervention trend had the intervention not occurred. It is equally – if not more - reasonable to assume that the trend would have followed the controls. The paper’s conclusion that an average of 2.5 days per employee was saved, both in the main text and in the abstract, needs to be tempered in the light of this. How best to do this will require some thought, which I will leave up to the authors.

P17. The authors state that: “We examined monthly averages of all-cause sick leave days and found no immediate intervention effect among OHS-based sick leave days but did detect a gradual intervention effect decreasing the number of OHS-based sick leave days..... The corresponding findings were similar in direction but not statistically significant for the control time series....”. However, table 3 shows that the results for the controls WERE statistically significant (p24 lines 11 through 19).

Minor points

P12. The authors included additional covariates intended to model the seasonal fluctuations in sick leave rates, using sin and cos functions. Can they support this with references to the method.

P13. Table 1 gives descriptive statistics for the OHS, non-OHS and “all” groups. The data looks odd: for example, the mean age for “all” (40.27) is lower than either OHS (44.67) or non-OHS (42.16). Even allowing for overlap between OHS and non-OHS groups I cannot believe that when combined, the mean age is lower than that of either group separately. In fact, for all variables the value for “all” is below both separate groups. I suspect that the columns have been confused and that the middle column contains the data for “all”.

A given individual can have multiple sick leave periods, and this implies that an OHS-prescribed sick leave can be subsequently followed by a non-OHS sick leave. This creates a potential for a patient’s experience of OHS-prescribed sick leave to influence their future behaviour, for example to self-manage or to use self-certification. Some discussion is required of the likelihood that the significant change in trend in non-OHS sick-leave after the point of intervention could be due to this. For 2017, is there any data on

	what proportion of non-OHS patients previously had OHS sick-leave? Figure 1. More information is needed on exactly how the black lines of “smoothed averages” were derived. Are these simple moving averages (and if so, of how many points, and how weighted) or something more complex?
--	---

VERSION 1 – AUTHOR RESPONSE

Reviewer: 1

Dr. Kristina Bengtsson Bostrom, R&D Centre Skaraborg Primary Care

Comments to the Review checklist

POINT 1. The study design is not appropriate to study the effectiveness of the intervention i.e. to study causality as also the authors point out. Using register data not intended for use in research is a special challenge and this should be reflected in the description of the results. Further, there is a risk that individuals beginning in the intervention group, went over to the control group in order to get a sick-note. Finally, the controls are not appropriately described to ensure that they were comparable with the individuals in the intervention group.

OUR RESPONSE: Thank you for the comment. We agree that the study design is not the best possible to study causality as we point out in the paper. Please see our detailed response to Points 3 and 5. Regarding the questions about the control group, please see our responses to Points 3 and 31.

We also agree that using administrative registers has its limitations as the referee points out. However, in Finland as in other Nordic countries register studies are carried out in a quite large scale and the quality of the administrative registers is high in international comparison. In fact, very few registers are constructed originally for research purposes. We have now added this to the discussion section (p. 19) and write as follows:” The use of administrative registers in research always entails limitations as they are not typically collected for research purposes. Nevertheless, in general the quality of registers in Finland as in other Nordic countries is good and administrative registers offer unique opportunities and are frequently used in research”.

POINT 2. The method especially of the intervention is not enough described. For instance, how the guidelines were implemented and iterated to newly employed staff. How other staff apart from physicians were involved in the intervention? This should be described in more detail.

OUR RESPONSE: Thank you for pointing this out. As a part of the project, we carried out a qualitative interview study investigating the implementation of the intervention. A thorough description of the implementation of the intervention will be published in another paper (Horppu et al. manuscript). However, we have now added a more detailed description of the implementation of the intervention to the text (p.8):” Training regarding the implementation of the guidelines was organised in OHS for physicians and all other relevant health care professionals such as physiotherapists and nurses. Short educational sessions were offered, and an e-learning course and some coaching sessions were led by a pain specialist. An intensive follow-up of the sick leave trends was started in OHS. For instance, recent trends of sick leave were reported in the monthly meetings for the physicians. All new relevant OHS employees were briefed about the guidelines.”

POINT 3. The statistics is much elaborated. I am not an expert in this field so I recommend a statistician should look at this part. I think that there might be inherent weaknesses in the material that can't be compensated for by elaborated and complicated statistics.

OUR RESPONSE: A statistician was involved in selection of the methods based on opportunities and weaknesses of the data. Please find below a short description regarding the strengths and weaknesses of the design in relation to statistical methods.

Interrupted time series analysis (ITS), in contrast to simple pre-post designs (that compare single pre-intervention time point to a single post-intervention time point), use multiple pre-intervention and post-intervention observations, allowing the underlying trend to be accounted for. Confounding is rarely a problem in this analysis as population characteristics usually change only gradually in time. However, ITS cannot exclude other events (or interventions) occurring at the same time, and therefore we opted for a Controlled Interrupted Time Series Analysis (CITS). Thus, we could have looked merely at the series of sick leaves prescribed at the OHS (intervention was targeted at the physicians working at the OHS) but we opted to include a series of sick leaves prescribed elsewhere (no such intervention) as well.

We agree that our control group was not optimal (as discussed later). Consequently, in the CITS-analysis, we opted to analyse the trends of sick leave days prescribed at OHS (intervention group) and those prescribed elsewhere (comparison group), separately. And did not include these in the same statistical model.

References:

- Lopez Bernal, J., S. Cummins, and A. Gasparrini, Interrupted time series regression for the evaluation of public health interventions: a tutorial. *Int J Epidemiol*, 2017. 46(1): p. 348-355.
- Lopez Bernal, J., S. Cummins, and A. Gasparrini, The use of controls in interrupted time series studies of public health interventions. *Int J Epidemiol*, 2018. 47(6): p. 2082-2093.
- Degli Esposti, M., et al., Can synthetic controls improve causal inference in interrupted time series evaluations of public health interventions? *Int J Epidemiol*, 2020.

POINT 4. The y-axis in the panels of figures (1 and 2) have different scales, which makes it difficult to compare the results in the intervention and control groups. The time scales are also different in figure 2 panels a and b compared to panel c and d. In these figures it seems to be a rise in sick days the succeeding years 2017-2019. These data is not commented.

OUR RESPONSE: The point of figures was not so much to compare absolute values across intervention and comparison groups but rather to compare the change in values from pre- to after-intervention period within the panels, and to compare such relative trends across the panels. The absolute values for OHS vs. other prescribing services could differ for many reasons but a trend difference around the intervention indicates that the difference was due to the intervention. The absolute amount of sick leave from OHS vs. other sources differed to the extent that using exactly same scales across the figure panels would hamper the illustration of that trend. To highlight the key message (difference in trends), we chose to use different scales for the figure panels despite it being sub-optimal for comparing the absolute values.

We now have added the following text (p. 14): "It should be noted that the absolute values in the Panels of Figures 1 and 2 are not comparable to each other as Y-axes are different. The figures

illustrate how the trend in sick leave days changes at the time of the intervention rather than compare absolute values. Due to relatively small change in trend, similar Y-axis would not illustrate the trend. “

POINT 5. In the discussion the authors use the word effect to describe the presumed outcome of the intervention. I suggest that a less decisive word is used here as I think there is a risk for bias in the data that make the conclusions uncertain.

OUR RESPONSE: We agree that the causality remains unclear as we state in the conclusions. However, the estimation of the intervention effect is the aim of the paper and the general methodology in this quasi-experimental study. We must discuss the target of estimation at some points, while also discussing the quality of the estimate. This is different from discussing potential causal implications of e.g. correlation coefficient, because the target of estimation would be in that case association rather than effect. The title of the manuscript, abstract and the main text have been revised to further address this concern of the reviewer and we hope our use of language aligns appropriately with the intended concepts.

POINT 6. The language needs revision, and very long sentences in several places in the manuscript should be shortened.

OUR RESPONSE: Thank you for pointing this out. The language of the manuscript has been revised by a native English speaker.

POINT 7. Studies of interventions with the aim to reduce sick leave are much longed for but are also difficult to perform. The current study is from Finland where the authors have used register data to evaluate an intervention using guidelines on musculoskeletal disorders aimed at sick-listing physicians. I have several issues that need to be addressed.

Title The word effectiveness gives the impression of causality which I think is too strong as there are many shortcomings of the data, which also the authors recognize. It could be an association worth to investigate under more strict conditions. The wording in the title, abstract and main text of the manuscript should be revised with this in mind.

OUR RESPONSE: Thank you for your comment. Please see our response to Point 5.

POINT 8. Avoid references in Finnish language. For instance I recommend instead of reference #8 the authors might use: Kankaanpää AT, Franck JK, Tuominen RJ. Variations in primary care physicians' sick leave prescribing practices. *Eur J Public Health*. 2012;22:92-6. doi: 10.1093/eurpub/ckr031. PMID: 21441559?

Also, reference #9 might be replaced by Hinkka K, Niemelä M, Autti-Rämö I, et al. Physicians' experiences with sickness absence certification in Finland. *Scand J Public Health*. 2019;47(8):859–866.

Does reference #11 have an Abstract in English? If not, the results ought to be described in more detail in the text.

OUR RESPONSE: The references have been changed as suggested. Reference # 11 has an Abstract in English.

POINT 9. Methods. The nature of the intervention is not enough described. How were the guidelines implemented and iterated to newly employed staff? What did the guidelines look like? Were they including diagnostic procedures, treatment options in addition to the recommended sick leave length? Were other staff apart from sick leave prescribing physicians involved in the intervention for instance physiotherapist, occupational therapists, or psychologists treating the patients? Were also the employers involved in the intervention? Especially the practical instrument used in this context (page 7, line 51) and the checklist described on page 9, line 3 should be described or attached in an appendix.

The control group described on page 9, line 9 includes individuals with self-certification. I presume that the controls thus include individuals with common cold and other self-limiting conditions that could distort the results. Why didn't you exclude those with self-certification from the study?

OUR RESPONSE: Thank you for the comment. Regarding the implementation of the intervention, please see our response to Point 2. The guidelines for low back pain (as an example) have now been translated into English and have been attached as an appendix (Appendix 1) (with the permission of the organisation), as suggested.

Unfortunately, we did not have data on self-certification of sick leaves available in these analyses. However, self-certification of sick leaves concerns short-term sick leaves (max 3 to 5 days) only and is used only to a minor extent. This is mentioned in the manuscript (p.5) as follows: "The employees of the City of Helsinki can use self-certification for sick leaves lasting up to five days. In the case of self-certification, a medical certificate is not needed, but the employee notifies the supervisor when taken ill. In 2016, self-certification was mostly used among young employees (under 30 years of age) (Sumanen et al. 2018)."

POINT 10. Outcome, page 9. Were any patients sick listed part-time? If so was this accounted for in the analysis?

OUR RESPONSE: Thank you for the question. Unfortunately, we did not have information on part time sick leaves and for that reason we could not take this into account in the analyses. A sick leave day was counted as a sick leave day. In Finland, partial sickness benefit can be granted only after 11 days of full-time sickness absence. Partial sickness benefit days comprised only 7% of all compensated sickness benefit days in Finland in 2016. This has now been added to Discussion (p. 20).

POINT 11. Covariates, page 9. Among diseases that were included as covariates in the analyses depressive disorders are not mentioned. I would have expected that depression, burn-out syndrome and other psychiatric conditions should be included. These are conditions associated with chronic pain and might make return to work difficult. Are psychotropic drugs a proxy for these conditions and if so were the underlying diagnoses validated and medication prescriptions occurring simultaneously with the sick leave periods?

OUR RESPONSE: Data on purchases of psychotropic drugs was indeed utilised as a proxy for these conditions. The data were drawn from the national registers of the Social Insurance Institution of Finland (p. 6) and as such diagnoses were validated. The timings of purchases were described as follows (p.9): "Covariates that were available in the registers and were regarded as potential confounders included age, sex, occupational class

(based on occupational title and categorised as upper grade non-manualemployees, intermediate grade non-manual employees, lower grade non-manual employees and manual workers), job contract (permanent, temporary) valid 1st January 2016, chronic somatic illnesses (derived from the Register of Special Reimbursement for Medication Purchases, and included diabetes, heart disease, rheumatoid arthritis, chronic asthma, stage 2 hypertension, Parkinson's disease, epilepsy, uremia, bowel disease, multiple sclerosis disease, and diseases of pancreas and categorized as 0=no, 1=yes) valid 1st January 2016, and purchases of prescribed psychotropic medication (Anatomic Therapeutic Chemical, ATC classification N06 or N05 categorised as 0=no, 1=yes) between 1st November 2015 and 1st November 2016. "

POINT 12. Statistical analyses. The methods are extensively described. I suggest that a great part of the text, specifically on page 11, could omitted or moved to an appendix.

OUR RESPONSE: We considered moving most of the technical material to a supplement, but eventually opted against it, as we have a request to expand the technical material from the reviewer #4 and the article is quite technical in nature. Clearly, there are arguments for both including this information into the main text and for adding it to supplemental material. We wish the editors decide which way is preferable.

POINT 13. Page 12, line 10-19. I don't understand how the authors handled individuals with a mixture of sick leave certificates from OHS and other sources. Do I perhaps misunderstand or does this mean that a single individual could be in both groups, in separate time intervals? If so, how could the authors disentangle the effect of the intervention from the "control effect"? I suggest exclusion of individuals with a mix of certificates from OHS and control, especially if the "mix" occurred during a short interval of time.

OUR RESPONSE: Thank you for the question. The reviewer has correctly understood that a same individual could seek a sick-leave certificate either from OHS or from other health services (e.g. public sector). At different times, he or she may have approached a different service. This is precisely the reason why we could not apply the usual CITS model over all the data, but instead examined separate intervention and comparison groups of outcome values. The intervention of interest was applied to the health care staff at OHS and therefore its effects are expected to show in the sick leaves (outcome values) prescribed at OHS. We have now clarified this aspect of the data in Methods (p. 12) as follows: "Because our outcome data timepoints (months) could include sick leave days from prescribed at OHS and other sources, defining a covariate for intervention group membership would have required us to prioritise either OHS or other sources. It is typical in CITS design to include a binary covariate that gets a value 1 when a unit (here a person) belongs to the intervention group and value 0 when in the control group. In this study, participants had sick leaves both from OHS and non-OHS sources. Instead of formal testing of such covariate, we conducted a separate ITS-design and analyses for the intervention and comparison groups."

Finally, we noted that "Of those employees who were prescribed sick leave at OHS, 95% were prescribed sick leave also elsewhere (or they used self-certification)" (page 13). This feature of the data (and local health-care systems) made it difficult to carry out the reviewer's suggestion in practice.

POINT 14. Results.

Page 13, line 6, "87% of all employees were prescribed sick leave at least once" is in contrast to line 18; "Of all employees 30 % were not on sick leave at all". Please explain this discrepancy.

OUR RESPONSE: Thank you for noticing this mistake we have made in the manuscript. The correct figure is 70%. This has now been corrected in the manuscript. (p. 13).

POINT 15. Data in Table 1 is unclear. It seems that 1,412 individuals, ($1,412 = (25,200 + 52,174) - 75,962$) were prescribed sick leave both at OHS and by others i.e. they belong to both groups making the comparison between groups difficult. Further, how come that the mean age is lower in “All” compared to “sick listing at OHS” and “other sick leaves”? Is this because the 1,412 individuals are counted twice?

OUR RESPONSE: Thank you for this question. Please note that the footnote emphasises how “many employees are prescribed sick leave at OHS and elsewhere”. As ~95% of employees with OHS sick leaves also had other sick leaves, much more than 1,412 individuals overlap in the groups (i.e., ~23,940 in the OHS group). The most likely reason that age is lower in the total population than in those with either kind of sick leave is that older employees are more likely to get prescribed sick leave on average. That is, the group “All” contains many employees (~30%) without any kind of sick leaves during our follow-up period. We added a further footnote referring to the column title “All” to stress the column also contains individuals not in either of the two groups.

POINT 16. Figure 1-2. The scales on the x-axis are different in the panels describing the results in the intervention and control groups, which can distort the impression of change in sick leave days and periods. The presumed effect of the intervention also seems to disappear in later years (Figure 1) which should be discussed by the authors.

OUR RESPONSE:

Thank you for the observation. Unfortunately, the register we used to distinguish OHS sick leaves from other sick leaves was sampled for a shorter time than the employer’s register containing the employee-recorded sickness absence. By looking at all sick leaves without distinguishing OHS from other sources, we could examine a longer time series that contains also the OHS sick leaves and could reveal their effect for a longer time than the OHS-only series. For this reason, we specifically wanted to assess all the three series: OHS, others, and any source.

POINT 17. Page 13, lines 53 and following; the interpretation of the results belongs to the Discussion section.

OUR RESPONSE: Thank you for the comment. The text has been modified as follows (p.13–14): “Based on a visual inspection of monthly averages, the number of all sick leave days prescribed at OHS declined shortly after the intervention had started in 2016 (Figure 1a) and similar temporal changes were not observed for other sick leave periods (Figure 1b). After 21st April 2017, our register data could not differentiate between sick leave days that were OHS-based and other sick leave days, but we had a longer stretch of data when counting in all sick leaves. A decline in sickness absence after intervention was seen in a visual inspection of all sick leaves (thick line for local-regression smoother), as well as annual periodic variation (thin dashed line in Figure 1c). When removing cyclic annual variation and a linear trend, the visual suggestion of an intervention effect remained. However, sick leaves appeared to return to preceding rates towards the end of the longest available follow-up data (in 2018) (Figure 1d).”

POINT 18. Page 15, lines 53 and following. The calculation of the reductions in costs due to the estimated reduction in sick leave days in the intervention group is not supported by any observed

data. In a defined population i.e. all employed in the City of Helsinki there ought to be data on reimbursement costs to support the calculations in this study. I suggest that the authors include such data.

OUR RESPONSE: Thank you for this comment. By “reimbursement costs”, we presume the reviewer refers to our hypothetical example of “A company with 40,000 employees and a direct cost of approximately 250 EUR per sick-leave day (numbers similar to those of the City of Helsinki)”. We have now further added the word “hypothetical” before “company” (p.17) to emphasize that this is an illustrative calculation in which the reader can place his or her own figures instead of ours. At the same time, the figures are taken from the Statistics of Finland and Salmela et al. 2020. We believe giving this kind of back-of-envelope calculation is appreciated by the readers trying to grasp the practical consequences of the results but, of course, are willing to remove it if considered inappropriate by the editor. This part has now been presented in the Discussion (p. 17).

References:

- Statistics Finland. (2018, October 15). Official statistics of Finland: Index of wage and salary earnings 2018, 3rd quarter. Web publication. Retrieved from: https://www.stat.fi/til/ati/2018/03/index_en.html

- Salmela J, Lahti J, Mauramo E, Pietiläinen O, Rahkonen O & Kanerva N (2020) Associations of changes in diet and leisure-time physical activity with employer’s direct cost of short-term sickness absence, *European Journal of Sport Science*, 20:2, 240-248, DOI: 10.1080/17461391.2019.1647289

POINT 19. Discussion. The authors state that there was “a clear gradual intervention effect decreasing the number of OHS-based all-cause sick leave days. Similar gradual decreasing trend was found in the control time series, but that was offset by an increase in the sick leave days and periods at the time of intervention”. The sharp rise of sick leave days in the control group immediately after the intervention is surprising. As the author speculate it could be due to patients moving from the intervention group to the control group in order to be sick listed. This is a very plausible explanation in my view. Thus, the comparison between these groups might not be a valid measure of effects of the intervention. Especially the notion that the decline in sick leave days in the control group was off-set by the rise in the beginning. This could also be interpreted as an off-set to the decline in the intervention group (as patients from OHS could seek a second opinion of their need for sick leave in caregivers outside OHS). Somehow, the suspected spillover of patients from the intervention to the control group must be investigated in the data and accounted for in the analysis. Further, other data for instance changes in the costs (savings of 25 million Euros) for sick leave ought to be included to support the authors’ conclusions.

OUR RESPONSE:

We have now replaced the term “control group” by “comparison group” throughout, and in the above comments we explained how ITS could be applied also without any sort of a comparison. Thus, our analysis should be thought of as being in between ITS and CITS in what comes to scientific value of evidence. This has now been added to p. 19 as follows:” However, only the OHS professionals were exposed to the intervention and the time series for the intervention and comparison groups were analysed separately. Thus, our analysis should be thought of as being in between ITS and CITS in what comes to scientific value of evidence.”

If we could have formed a proper control group, we would have used CITS and the group status would have been a covariate only, not a separate ITS analysis. Please see Point 18 regarding the question of costs of sick leaves.

POINT 20. The authors acknowledge this situation, page 18, lines 34 and following. The conclusions

should accordingly be much more cautious. Of course, as the authors state a prospective intervention study that would enable tracking the patients' sick-listing behavior could shed further light on the effectiveness of such an intervention.

OUR RESPONSE: The conclusions now read as follows (p.20): "A gradual change of the trend at the time of the intervention was observed, resulting in a decline of the number of sick leave days and periods prescribed in OHS. This is consistent with the intervention being effective, but the causality of the relationship is unclear due to similar findings for the comparison group".

Reviewer: 2

Dr. Priti Mulimani, University of Washington

Comments to the Author:

POINT 21. As a health-care practitioner in an international setting, I have no exposure to the Finnish health system, but this paper for the most part does a pretty great job of explaining the functioning and administrative aspects underlying the Finnish health system study in a way that is understandable for international readers. For the most part this is a very well-planned, -executed and -written study. I had only the following few suggestions / questions

OUR RESPONSE: Thank you for the positive feedback, it is highly appreciated. We have tried our best to take the suggestions into account in the revised manuscript.

POINT 22. 1. From reading the paper I inferred that "sick-listing" refers to the practice of recommending sick-leave to the patients by the health care provider. It would be helpful to have a definition/explanation of the term "sick-listing" in the beginning, for uninitiated readers.

OUR RESPONSE: Yes, sick listing is used as a synonym for prescribing sick leave. This is now explained on p.4 as follows:" Furthermore, earlier studies have found that sick listing practices (practices of prescribing sick leave) vary a lot across physicians."

POINT 23. The control time-series included patients with "no appointment at OHS, but appointment and sick-listing had taken place elsewhere or the employee had used self-certification, in which case there had been no medical appointment at all". How did the study account for self-medication (previously purchased, hence no current purchase registered in the pharmacy) or other self-interventions in this group which would have gone unreported?

OUR RESPONSE: The term self-certification refers to a practice of taking sick leave without a need for medical appointment (and certificate for sick leave). Please also see Point 9. As we utilised large scale administrative register data , we did not have access to data on self-medication (this would demand access to self-reported information, e.g. survey data).

POINT 24. In methodology please provide a reference for – "Analyses were run separately for men and women (who are well known to differ in sick leave behaviour)..."

OUR RESPONSE: Thank you for pointing this out. As the sick leave literature is substantial and we already have a lack of space in the manuscript, this text has now been deleted from the manuscript (p.12).

POINT 25. Comparison with previous studies and their findings has been broadly summarized in a paragraph. I would prefer to see more granular details as to how specific findings from the current study fared with respect to the previous one

OUR RESPONSE: To our best knowledge, this is the first study to explore quantitatively the effects of introducing guidelines for prescribing sick leaves for musculoskeletal disorders. Some studies exist that compare different methods of implementation, but they are not comparable to our study. The literature that evaluates various interventions that attempt to change the behaviour of physicians in other clinical areas is vast, but not comparable to our study. We mention this now on p.18:

“As far as we know, there are no previous studies investigating the effects of introducing guidelines on prescribing sick leaves (as compared to the situation where no guidelines exist). There are a few studies that compare different ways of implementation to each other (Becker et al. 2008; Rossignol et al. 2000). There are a great number of previous studies on interventions attempting to change physicians’ behaviour in other areas of clinical practice. Findings show that success in changing the behaviour and adherence to different clinical guidelines varies. Some studies have reported success in behaviour change [20, 21] whereas other studies have reported limited success or found interventions ineffective [22-24]. Moreover, a change in physician’s behaviour does not automatically reflect in patient outcomes [23]. Guidelines have been found to be a necessary but an insufficient step in changing clinical care [25]. The importance of proactive implementation of guidelines and efforts to include the new practice into existing organisational procedures, which both were attempted in this intervention, have been emphasised to achieve sustainable results [26, 27]. As the context of a complex intervention is important, the results of an intervention in one environment cannot directly be applied to another context.”

POINT 26. “We modelled the number of sick leave days using binomial regression. “Binomial regression” implies that the outcome was a binomial (i.e. could take 2 values only): this will be confusing to many readers. It gives the impression that the outcome has been dichotomised, when it has not.

OUR RESPONSE: Thank you for the comment. “Binomial regression” refers to a binomial outcome distribution, which is different from binary outcome. Binomial and negative binomial regression models are often used when outcome is count data. We agree the distinction is a somewhat subtle one, as number of sick days in a month (in 30 days) can be expressed via 30 binary values of sick day vs. non-sick day. Since our data and covariates were at the level of months (30 days), we are convinced that binomial regression rather than logistic regression is the correct term here. As a reference we now cited (p.11) the classic book by Gelman and Hill (2007).

- Gelman, A., & Hill, J. (2007). *Data Analysis Using Regression and Multilevel/Hierarchical Models*. Cambridge University Press.

Reviewer: 3

Dr. Takeshi Toyooka, Nishikawa Orthopaedics Clinic

Comments to the Author:

POINT 27. Thank you for your great work. Although the article have important message, there are too much results in your article.

Which factor will affect to the intervention effect?

I think there are no need to show without table data.

The content of the cost is in the results, but this should be moved to the discussion section. As you know, if you want to compare 4 group's average data, I think two-way ANOVA is more appropriate. Don't you need to compare "sick listing at OHS" group and "other sick leaves" group? Otherwise, would you divide the purpose into intervention effect and its duration?

OUR RESPONSE: Thank you for the positive feedback. The recommendations for prescribing sick leave was the main component of the intervention (p.7,20):" The recommendations for physicians constituted chronologically the first and the main element in the intervention. Direct access to physiotherapists without a doctor's referral was introduced in the following year, and thereafter it was not possible to differentiate the effects of these different elements." (p.20).

We have now moved some of the text which previously was in results to discussion section, including the text on financial implications.

We only compare two groups, intervention group (sick listing at OHS) to comparison group (other sick leaves). There is overlap between groups, and the comparison group is suboptimal, but best that we could achieve. Please also see our responses to POINT 3 regarding the comparison group.

Reviewer: 4

Dr. David Reeves, University of Manchester

Comments to the Author:

POINT 28. This paper applies an interrupted time-series design to try and assess whether an intervention consisting of recommendations on pain management and sick leave prescribing for OHS physicians, reduced numbers of sick leave days taken by employees of the City of Helsinki, Finland over the subsequent one-year period. The paper is interesting and the analysis mostly sound, however there are some important weaknesses that need to be addressed before publication could be recommended, as described below.

OUR RESPONSE: Thank you for the positive feedback. We have tried our best to take the suggestions into account in the revised manuscript.

POINT 29. Some justification, supported by references, for using a binomial model is required. The concern is that multiple models may have been fitted and the binomial chosen on the grounds that it gave the most favourable results – a good justification is therefore required to dispel such a concern. In addition, because of the above concerns, sensitivity analysis is required to assess the robustness of the results against distributional assumptions. My recommendation would be to run at least two additional sensitivities: (1) using a Negative Binomial model; (2) using boot-strapped SEs that make no distributional assumptions. The authors might also want to consider using a zero-inflated negative binomial model, though interpretation may get very complex in the context of interrupted time-series.

OUR RESPONSE: Thank you for this comment. Negative Binomial distribution is typically considered an alternative to the Poisson distribution as both are defined on all non-negative integers, up to infinity. Negative binomial regression is for modeling count variables, particularly for over-dispersed count outcome variables. Our outcome was number of SA days in 30 days, which has a clear upper limit for the support of the distribution, and the distribution is highly skewed. We have added a standard reference to the manuscript (p.11) as follows: "Given the data characteristics above, the outcome was modeled using generalised (binomial) linear regression model and the models were estimated with generalised estimating equations (Gelman and Hill, 2007, page 112)."

- Gelman, A., & Hill, J. (2007). Data Analysis Using Regression and Multilevel/Hierarchical Models.

Cambridge University Press.

We also avoided bootstrapping of standard errors: while they theoretically could be more accurate with sufficient resampling, our GEE estimates in data this large took a prohibitively long amount of time for replicating over 2000 times (we lacked access to a high-performance computing cluster for privacy-sensitive data). Specifically, one estimation took ~4.5 min. Thus, 2000 bootstrap replications of the 12 models would take roughly 75 days provided no convergence errors occur in any of the 24000 model estimations. This would be excessive given there is no reason to expect asymmetric confidence intervals for regression coefficients and given that standard Wald intervals are typically reported in epidemiology (i.e., bootstrapping tends to be reserved for cases where estimation is difficult for some reason).

POINT 30. No formal statistical comparison is made of the regression parameters for the OHS and control groups, even though this should be relatively easy by including both groups in the same regression and interacting the regression terms with a group identifier. Since a direct comparison would normally be expected, some justification for this decision needs to be given. Personally, I have some qualms over the control group as I am not convinced that they represent a reasonable counterfactual: first, table 1 shows that a much smaller proportion of controls were on permanent job contracts, and second – probably related – prior to the intervention, mean numbers of sick-leave days for controls were more than twice those for the OHS-group – this suggests that pre-intervention, OHS may have already been operating to a very different protocol compared to “other” services. Thus I would suggest that there are important systematic differences between the groups making a direct comparison problematic.

OUR RESPONSE: Thank you for the comment. Indeed, we have now renamed the control group as a “comparison group” to highlight that it is not a counterfactual (please see POINTS 3 and 19). We have justified the lack of direct comparison via group identifier with the fact that ~95% of the OHS group were also in the other sick-leaves group. That is, there are no distinct groups of individuals from OHS vs. elsewhere but distinct sources of sick-leave prescriptions.

We have added the following discussion under limitations, p. 19:

“The validity of the comparison group can be debated [17] as all employees could be prescribed sick leave both at OHS and elsewhere (or use self-certification in short sick leaves). However, only the OHS professionals were exposed to the intervention and the time series for the intervention and control groups were analysed separately. Thus, our analysis should be thought of as being in between ITS and CITS in what comes to scientific value of evidence.”

POINT 31. Early on in the paper the authors make a lot of the claim that they included a control group. However, controls are only informative to the degree that they are similar enough to the intervention group to make comparisons meaningful. In this study the controls occupy an awkward middle-ground: they are not similar enough to justify formal comparisons (in my opinion), but may provide useful contextual information. I think use of the term “controls” here is not advisable, since it implies a high degree of comparability. The authors should consider using the term “comparison group” instead, and give suitable caveats – such as those I have expressed above – as to why they should not be treated as direct controls.

OUR RESPONSE: We agree, and would have liked to use CITS (Controlled Interrupted Time Series)-analysis with direct controls, but could not because a same individual could have had sick-leaves (outcome values) from both the sources: OHS and others (please see also our responses to other points). Due to this limitation in the data, we could only compare findings for the OHS counts and

other counts. As suggested, we have replaced the term control group by comparison group throughout the paper.

POINT 32. In two respects the seasonal sin/cos wave is not a particularly good fit to some of the data. First, from Figure 1 and 2 it can be seen that there is a strong “Christmas effect” on sick leave, with the December(?) value dropping right down in just about every year – perhaps because many people were taking holiday? These points strongly diverge from the wave function. Second, the wave is a particularly poor fit to the ONS group (figure 2a), with the large majority of points in the pre-intervention period being very high above the wave. This suggests that the fit of the pre-intervention slope is highly biased, supported by the fact that it also deviates greatly from the moving average line in figure 1a. Over the pre-intervention period I counted 27 data points above the pre-intervention line of slope but only 10 below it: this is strongly suggestive of a bias. The fit is also poor to the post-intervention period. I note in passing that the same problem does not seem to affect the model fit to the control dataset (figure 2b), though the “Christmas effect” is still very evident.

OUR RESPONSE: Thank you for this comment. We considered it more parsimonious to model annual cycles with two covariates instead of 11 or 12. The number of months per covariate would get low with that many extra covariates, strongly risking overfitting. As for Figure 2a, we have clarified the text in the figure caption: “Observed data points denote to simple monthly averages and the lines represent GEE-model fits to employee-level data”. We considered omitting the points from this panel, as the model was not fit for them but decided to leave them as a partial indicator of raw data and for consistency. The data points could be removed from the panels 2a and 2b, if the reviewers and the editor consider it better. With the periods data, the model was fit to the data points shown, which explains its better fit.

POINT 33. Given this, I think it essential that the OHS group model be re-evaluated. If figure 2a is accurately drawn then something has clearly gone wrong in the analysis. One option might be to test the sensitivity to the sin/cos wave assumption, by replacing it with calendar month (jan, feb, etc) as a categorical variable; this would also resolve the Christmas effect issue.

OUR RESPONSE: Please see the above comment on figure 2a not showing modeled data but monthly averages. Modeled data would be difficult to plot. Please see also our response to above point regarding the monthly covariates.

POINT 34. The authors estimate the mean 12-month reduction in sick leave days resulting from the intervention. The calculation is mathematically correct. However, it is based on assuming that the observed trend prior to the intervention would have continued unchanged had the intervention not happened. This is where I have a real difficulty with the paper. The control time-series shows a similar (and significant) change in trend after the intervention. Comparisons are made complex by the “jump” in mean sick days for the control group at the start of 2016, but even allowing for this, the mean number of sick days for controls at the end of 2016 was no greater than at the start.

Thus on the evidence of the control group, it is not valid to assume that sick leave in the OHS group would have continued along the pre-intervention trend had the intervention not occurred. It is equally – if not more - reasonable to assume that the trend would have followed the controls. The paper’s conclusion that an average of 2.5 days per employee was saved, both in the main text and in the abstract, needs to be tempered in the light of this. How best to do this will require some thought, which I will leave up to the authors.

OUR RESPONSE: Thank you for this remark. We designed the referred calculation mainly to highlight

the practical implication of the regression coefficient, and then indeed discussed the limitations the reviewer brought up. We are not sure whether the reviewer refers to “pre-intervention trends” in the monthly population averages or those in the GEE estimates based on individual-level data. The discrepancy in these are striking, and one of the reasons we separately reported results for monthly average vs. employee-level data. We suspect the differences between monthly-average- and GEE-models are related to under-dispersion and within-individual correlated errors in the individual-level data, both of which the GEEs take into account.

It is easy to subtract the comparison group trend to give a more toned-down estimate but the limitations of the comparison group stand, of course. We do not have a perfect solution to this issue but we have now added also this toned down estimate to the manuscript as follows (p.17).

“If one assumes that the trend in OHS sick leaves would have followed changes in the comparison group instead of the pre-intervention trend, one could subtract the gradual effect in comparison group from that of the OHS group to derive a conservative OR estimate ($0.994 = \exp(-0.014+0.008)$, from Table 3 coefficients) and then repeat the calculation. This would yield 1.01 avoided sick-leave days per year per employee, with estimated approximate savings of 10.1 million EUR per year.”

Please, notice that the small standard errors of the coefficients ensure that such joint estimates are not meaningless.

POINT 35. The authors state that: “We examined monthly averages of all-cause sick leave days and found no immediate intervention effect among OHS-based sick leave days but did detect a gradual intervention effect decreasing the number of OHS-based sick leave days..... The corresponding findings were similar in direction but not statistically significant for the control time series....”. However, table 3 shows that the results for the controls WERE statistically significant (p24 lines 11 through 19).

OUR RESPONSE: As per the citation of the reviewer, the non-significant findings pertained to results about the monthly-average data, whereas the Table 3 pertained to the GEE models of employee-level data. Indeed, these results differed for the comparison group, as discussed in our response to the comment immediately above.

POINT 36. The authors included additional covariates intended to model the seasonal fluctuations in sick leave rates, using sin and cos functions. Can they support this with references to the method.

OUR RESPONSE: We have now included a citation to a classic book in time-series modeling as follows (p. 11): “Besides this usual ITS setting, we took into account the annual fluctuation of sick leave by modelling an annual trend estimating coefficients for two additional covariates with values $\sin(m2\pi/12)$ and $\cos(m2\pi/12)$, where the factor $2\pi/12$ scales monthly data points m to the annual cycles (Shumway & Stoffer, 2017, page 64).”

- Shumway, R. H., & Stoffer, D. S. (2017). Time Series Analysis and Its Applications: With R Examples (4th ed.). Springer.

POINT 37. Table 1 gives descriptive statistics for the OHS, non-OHS and “all” groups. The data looks odd: for example, the mean age for “all” (40.27) is lower than either OHS (44.67) or non-OHS (42.16). Even allowing for overlap between OHS and non-OHS groups I cannot believe that when combined, the mean age is lower than that of either group separately. In fact, for all variables the value for “all” is

below both separate groups. I suspect that the columns have been confused and that the middle column contains the data for “all”.

OUR RESPONSE: Thank you for the observation. We added a further footnote to the column title “All” to stress the column contains also individuals not in either of the two groups (Please see our response to Point 15). The mean age is lower in the total population than in those with either kinds of sick leaves because older employees are more likely to be on sick leave . That is, the group “All” contains many employees (~30%) without any kind of sick leaves during our follow-up period.

POINT 38. A given individual can have multiple sick leave periods, and this implies that an OHS-prescribed sick leave can be subsequently followed by a non-OHS sick leave. This creates a potential for a patient’s experience of OHS-prescribed sick leave to influence their future behaviour, for example to self-manage or to use self-certification. Some discussion is required of the likelihood that the significant change in trend in non-OHS sick-leave after the point of intervention could be due to this. For 2017, is there any data on what proportion of non-OHS patients previously had OHS sick-leave?

OUR RESPONSE: Thank you for pointing this out, we agree. This is discussed in page 18 of the manuscript as follows:

“Nevertheless, the findings may also suggest that soon after the intervention carried out in OHS, the employees increasingly sought treatment elsewhere (or used self-certification increasingly) as a rise in the level of other sick leaves was detected. This trend leveled down, however.”

And page 19 as follows:

“The validity of the comparison group can be debated [17] as all employees could be prescribed sick leave both at OHS and elsewhere (or use self-certification in short sick leaves).”

In page 13 we mention that “Of those, who were prescribed sick leave elsewhere (or used self-certification) at least once, 46% were prescribed sick leave also at OHS.”

POINT 39. Figure 1. More information is needed on exactly how the black lines of “smoothed averages” were derived. Are these simple moving averages (and if so, of how many points, and how weighted) or something more complex?

OUR RESPONSE: Thank you for pointing this out. The following text was added to the Methods section (p.12): “Local regression lines for figures were drawn using loess-function of “stats” R package, version 3.5.1, with the default options.”

Reference: W. S. Cleveland, E. Grosse and W. M. Shyu (1992) Local regression models. Chapter 8 of Statistical Models in S eds J.M. Chambers and T.J. Hastie, Wadsworth & Brooks/Cole.

VERSION 2 – REVIEW

REVIEWER	Bengtsson Bostrom, Kristina R&D Centre Skaraborg Primary Care
REVIEW RETURNED	22-Jul-2021

GENERAL COMMENTS	Ms. bmjopen-2020-047018_R1 entitled “An intervention at physicians’ treatment of musculoskeletal disorders and sickness certification: An interrupted time series analysis”. Comments to the Review checklist The authors have accepted most parts of my suggestions for the manuscript. Despite this I still have concerns about the methods used in the study. I think that the authors don’t take into consideration the draw-backs of the register data and mix of individuals in the groups used in the analyses. POINT 1. The authors have now changed the title of the manuscript and omitted the word effectiveness. Regardless of the quality of administrative registers used for other purposes than research they cannot in general be used for studies of effectiveness. They can generate hypotheses to be tested in future trials. The use of such registries should render cautious interpretation of the results. The word effective turns up in several places in the manuscript; in the result section of the abstract and in the main text. I think the words “effect, effective” related to the results in this study should be replaced with words that states that the results are associated to the intervention to make clear that the study is observational. The added text in discussion on page 19 is appropriate. But in the Article summary (Strength and limitation) the uncertainty of movement of individuals between the intervention and comparison groups should be stated as a limitation. POINTS 3, 13 and 15. I have still doubts concerning the comparisons between the groups. As the individuals have different characteristics (Table 1) the groups seem to be different in some respects. The legend to Table 1 states that many employees were in both groups. I don’t understand why these groups’ characteristics are compared as the groups comprise the same individuals from time to time. The group “All” in Table 1 includes individuals not prescribed sick leave in the study period. This is confusing. Why did you include these individuals in Table 1 as they are not included in the analyses? “All” could be misinterpreted as the two groups (OHS and “other sick leave”) were merged. According to the authors the comparisons were performed between the ITS and CITS and the individuals on sick leave could be in either group at different times (OHS group ca 95% in the “other sick leave” group). This is difficult to comprehend in relation to the comparison; it implies that sick leave days of the same individual could differ whether it was issued by a physician in the OHS group or the other sick-leave group. A patient in self-perceived need of a sick leave could choose to go to a more “generous” physician further complicates the interpretation of the results. If it had been a clinical trial this could be compared to dropout after randomization and a high dropout would compromise the validity of the result. This makes the word “effect” too strong, the observations from this study could at most be used as a hypothesis to test in a future trial.
--

	POINT 16. Figures 1-2. I propose that you add in the legend the different scales in the figures so that the readers are not misled by the visual impression of the difference. I don't understand what the authors mean by the "any source" it seems to be mix of the OHS and other "sick leave". How could the OHS change be tracked in the extended time when the sick leave came from "any source"? POINT 18. The reduction in sick leave days is transformed by the authors into monetary savings as a consequence of the intervention. As the intervention is in real life it should be possible to find reduction in the cost for sick leave benefits during the intervention time. Such data should strengthen the association of the intervention and reduction in sick leave days. If this is not possible I suggest that this calculation is omitted. POINT 19. Discussion The authors address the concerns in point 19 and state that the control group was renamed to comparison group. My concern was that regardless of whether the groups were compared or analyzed separately there was a sharp rise of sick leave days in the comparison group. This could be due to patients moving from the intervention group to the comparison group in order to be sick listed as also the authors suggest. As this could be suspected it diminishes a proposed effect of an intervention. This is also a reason to interpret the results cautiously. The conclusions should accordingly be much more cautious. Of course, as the authors state a prospective intervention study that would enable tracking the patients' sick-listing behavior could shed further light on the effectiveness of an intervention such as the one in this study.
--	---

VERSION 2 – AUTHOR RESPONSE

Reviewer: 1

Dr. Kristina Bengtsson Bostrom, R&D Centre Skaraborg Primary Care Comments to the Author:

Ms. bmjopen-2020-047018_R1 entitled "An intervention at physicians' treatment of musculoskeletal disorders and sickness certification: An interrupted time series analysis".

Comments to the Review checklist

The authors have accepted most parts of my suggestions for the manuscript. Despite this I still have concerns about the methods used in the study. I think that the authors don't take into consideration the draw-backs of the register data and mix of individuals in the groups used in the analyses.

POINT 1. The authors have now changed the title of the manuscript and omitted the word effectiveness. Regardless of the quality of administrative registers used for other purposes than research they cannot in general be used for studies of effectiveness. They can generate hypotheses to be tested in future trials. The use of such registries should render cautious interpretation of the results. The word effective turns up in several places in the manuscript; in the result section of the abstract and in the main text. I think the words "effect, effective" related to the results in this study should be replaced with words that states that the results are associated to the intervention to make clear that the study is observational. The added text in discussion on page 19 is appropriate. But in the Article summary (Strength and limitation) the uncertainty of movement of individuals between the intervention and comparison groups should be stated as a limitation.

OUR RESPONSE: Thank you for the comment. We have replaced the term “effect” with the term “association” wherever possible as suggested by the reviewer. We now also use the term “intervention estimate” and expressions like “immediate/graduate change in trend”.

We agree that our control group was not optimal in this study. This limitation is now included in the “limitations” part of the Article Summary. Therefore, in the CITS-analysis, we opted to analyse the trends of sick leave days prescribed at OHS (intervention group) and those prescribed elsewhere (comparison group), separately. We did not include these in the same statistical model.

However, we want to also bring to the attention that population-based register data has indeed been previously used for observational studies of effectiveness in several so-called quasi-experiments (please see examples of references below). Quasi-experimental design is often used when randomization is either ethically or practically impossible, such as in our study. In these cases, using proper study setting and proper statistical methods is essential.

A *quasi-experiment* is an empirical interventional study used to estimate the causal impact of an intervention on target population without random assignment. Quasi-experimental research shares similarities with the traditional experimental design or randomized controlled trial, but it specifically lacks the element of random assignment to treatment or control. Instead, quasi-experimental designs typically allow the researcher to control the assignment to the treatment condition but using some criterion other than random assignment.

There are several types of quasi-experimental designs, each with different strengths, weaknesses and applications. These designs include (but are not limited to):

- Difference in differences (pre-post with comparison)
- Nonequivalent control groups design
- No-treatment control group designs
- Nonequivalent dependent variables designs
- Removed treatment group designs
- Repeated treatment designs
- Reversed treatment nonequivalent control groups designs
- Cohort designs
- Post-test only designs
- Regression continuity designs
- Regression discontinuity design
- Case-control design
- Time-series designs
- Multiple time series design
- **Interrupted time series design**
- Propensity score matching
- Instrumental variables

*) Kausto J, Viikari-Juntura E, Virta LJ, et al. Effectiveness of new legislation on partial sickness benefit on work participation: a quasi-experiment in Finland *BMJ Open* 2014;**4**:e006685. <https://bmjopen.bmj.com/content/4/12/e006685>

*) Leinonen T, Viikari-Juntura E, Husgafvel-Pursiainen K, Juvonen-Posti P, Laaksonen M, Solovieva S. The effectiveness of vocational rehabilitation on work participation: a propensity score matched analysis using nationwide register data. *Scand J Work Environ Health*. 2019 Nov 1;**45**(6):651-660. 10.5271/sjweh.3823. Epub 2019 Apr 12. PMID: 30977515. <https://pubmed.ncbi.nlm.nih.gov/30977515/>

*) Bygren M, Szulkin R. Using register data to estimate causal effects of interventions: An ex post synthetic control-group approach. *Scandinavian Journal of Public Health*. 2017;**45**(17_suppl):50-55. doi:[10.1177/1403494817702338](https://doi.org/10.1177/1403494817702338)

*) James A. Lopez Bernal, Antonio Gasparrini, Carlos M. Artundo, Martin McKee, The effect of the late 2000s financial crisis on suicides in Spain: an interrupted time-series analysis, *European Journal of Public Health*, Volume 23, Issue 5, October 2013, Pages 732–736. <https://doi.org/10.1093/eurpub/ckt083>

*) Niederkrotenthaler T, Sonneck G. Assessing the Impact of Media Guidelines for Reporting on Suicides in Austria: Interrupted time Series Analysis. *Australian & New Zealand Journal of Psychiatry*. 2007;41(5):419-428. <https://doi.org/10.1080/00048670701266680>

POINTS 3, 13 and 15. I have still doubts concerning the comparisons between the groups. As the individuals have different characteristics (Table 1) the groups seem to be different in some respects. The legend to Table 1 states that many employees were in both groups. I don't understand why these groups' characteristics are compared as the groups comprise the same individuals from time to time. The group "All" in Table 1 includes individuals not prescribed sick leave in the study period. This is confusing. Why did you include these individuals in Table 1 as they are not included in the analyses? "All" could be misinterpreted as the two groups (OHS and "other sick leave") were merged.

According to the authors the comparisons were performed between the ITS and CITS and the individuals on sick leave could be in either group at different times (OHS group ca 95% in the "other sick leave" group). This is difficult to comprehend in relation to the comparison; it implies that sick leave days of the same individual could differ whether it was issued by a physician in the OHS group or the other sick-leave group. A patient in self-perceived need of a sick leave could choose to go to a more "generous" physician further complicates the interpretation of the results. If it had been a clinical trial this could be compared to dropout after randomization and a high dropout would compromise the validity of the result.

This makes the word "effect" too strong, the observations from this study could at most be used as a hypothesis to test in a future trial.

OUR RESPONSE:

In Table 1, column "all" includes all individuals included in the data and analyses (n=75 962). Please note that in the crude models in Table 3 n=75 962 as well. In the footnote of Table 1 it is mentioned that "Note that many employees were prescribed sick leave both at OHS and elsewhere (or used self-certification). The group "All" contains also employees with no sick leaves." Please also note that the unit of analysis was a sick leave day or a sick leave period, not an individual.

As suggested by the reviewer, we now use the term "association" or other expressions instead of term "effect" throughout the paper, wherever possible.

POINT 16. Figures 1-2. I propose that you add in the legend the different scales in the figures so that the readers are not misled by the visual impression of the difference. I don't understand what the authors mean by the "any source" it seems to be mix of the OHS and other "sick leave". How could the OHS change be tracked in the extended time when the sick leave came from "any source"?

OUR RESPONSE: Thank you for the comment. We now mention the differing scales in the figure caption. Any source was indeed a sum of OHS and other sick leave, but we had this data for a longer period. Because sick leaves from any source were a sum of sick leaves from OHS and elsewhere, any change in OHS sick leaves not coupled by changes in non-OHS sick leaves, will result in a change of all sick leaves from any source. Let $a_2 > a_1$ and $b_1 = b_2$, where subscripts index time points. Then, necessarily $a_2 + b_2 > a_1 + b_1$. Therefore, all sick leaves in our data track OHS sick leaves in time to some extent. By ITS design, changes were expected *specifically* in OHS sick leaves at a certain time point, and therefore also in sums of sick leaves that contain the OHS sick leaves.

POINT 18.

The reduction in sick leave days is transformed by the authors into monetary savings as a consequence of the intervention. As the intervention is in real life it should be possible to find reduction in the cost for sick leave benefits during the intervention time. Such data should strengthen the association of the intervention and reduction in sick leave days. If this is not possible I suggest that this calculation is omitted.

OUR RESPONSE: Thank you for the comment. Unfortunately, we did not have access to data that would have permitted analyses for the economic evaluation. The evaluation provided in the manuscript was based on a hypothetical example and calculations that were based on existing statistical information. Now we have omitted the calculations from the manuscript as suggested. We only refer to possibly savings as follows (p.17):” A hypothetical organisation with 40,000 employees and an average direct cost 250 of EUR per sick-leave day (numbers similar to those of the City of Helsinki) [22], is likely to have resulted in substantial annual savings at the time of the intervention.”

POINT 19.

Discussion

The authors address the concerns in point 19 and state that the control group was renamed to comparison group. My concern was that regardless of whether the groups were compared or analyzed separately there was a sharp rise of sick leave days in the comparison group. This could be due to patients moving from the intervention group to the comparison group in order to be sick listed as also the authors suggest. As this could be suspected it diminishes a proposed effect of an intervention. This is also a reason to interpret the results cautiously.

The conclusions should accordingly be much more cautious. Of course, as the authors state a prospective intervention study that would enable tracking the patients’ sick-listing behavior could shed further light on the effectiveness of an intervention such as the one in this study.

OUR RESPONSE: We modified the conclusions and they read now as follows: “The intervention *may* have reduced employee sick leaves and therefore it is *possible* that it had led to direct cost savings. However, further evidence for causal inferences is needed.”

VERSION 3 – REVIEW

REVIEWER	Bengtsson Bostrom, Kristina R&D Centre Skaraborg Primary Care
REVIEW RETURNED	27-Sep-2021
GENERAL COMMENTS	The authors have accepted my suggestions. The conclusion now appropriately suites the design of the study. They authors have also answered my questions in a satisfying way. There are a few typos in the text. I am satisfied and have no further comments.